# Microphysical processes producing high ice water contents (HIWCs) in tropical convective clouds during the HAIC-HIWC field campaign: dominant role of secondary ice production

Yongjie Huang[1], Wei Wu[2], Greg M. McFarquhar[2, 3], Ming Xue[1, 3], Hugh Morrison[4], Jason Milbrandt[5], Alexei V. Korolev[6], Yachao Hu[2, 7], Zhipeng Qu[6], Mengistu Wolde[8], Cuong Nguyen[8], Alfons Schwarzenboeck[9], and Ivan Heckman[6]

[1]Center for Analysis and Prediction of Storms, University of Oklahoma, Norman, OK, USA
[2]Cooperative Institute for Severe and High-Impact Weather Research and Operations, University of Oklahoma, Norman, OK, USA
[3]School of Meteorology, University of Oklahoma, Norman, OK, USA
[4]Mesoscale and Microscale Meteorology Laboratory, National Center for Atmospheric Research, Boulder, CO, USA
[5]Environment and Climate Change Canada, Dorval, Quebec, Canada
[6]Environment and Climate Change Canada, Toronto, ON, Canada
[7]Department of Atmospheric and Oceanic Sciences, School of Physics, Peking University, Beijing, China
[8]National Research Council Canada, Ottawa, Canada
[9]Université Clermont Auvergne, CNRS, UMR 6016, Laboratoire de Météor Physique, Clermont-Ferrand, France

**Correspondence:** Yongjie Huang (Yongjie.Huang@ou.edu; huangynj@gmail.com) and Greg M. McFarquhar (mcfarq@ou.edu)

**Abstract.** High ice water content (HIWC) regions in tropical deep convective clouds, composed of high concentrations of small ice crystals, were not reproduced by Weather Research and Forecasting (WRF) model simulations at 1-km horizontal grid spacing using four different bulk microphysics schemes (i.e., the WRF single-moment 6-class microphysics scheme (WSM6), the Morrison scheme and the Predicted Particle Properties (P3) scheme with one- and two-ice options) for conditions encountered during the High Altitude Ice Crystals (HAIC)-HIWC experiment. Instead, overestimates of radar reflectivity and underestimates of ice number concentrations were realized. To explore formation mechanisms for large numbers of small ice crystals in tropical convection, a series of quasi-idealized WRF simulations varying the model resolution, aerosol profile, and representation of secondary ice production (SIP) processes are conducted based on an observed radiosonde released at Cayenne during the HAIC-HIWC field campaign. The P3 two-ice category configuration, which has two "free" ice categories to represent all ice-phase hydrometeors, is used. Regardless of the horizontal grid spacing or aerosol profile used, without including SIP processes the model produces total ice number concentrations about two orders of magnitude less than observed at $-10°C$ and about an order of magnitude less than observed at $-30°C$, but slightly overestimates the total ice number concentrations at $-45°C$. Three simulations including one of three SIP mechanisms separately (i.e., the Hallett-Mossop mechanism, fragmentation during ice–ice collisions, and shattering of freezing droplets) also do not replicate observed HIWCs, with the results of the simulation including shattering of freezing droplets most closely resembling the observations. The simulation including all three SIP processes produces HIWC regions at all temperature levels remarkably consistent with the observations in terms of ice number concentrations and radar reflectivity, which is not replicated using the original P3 two-ice category configuration.

This simulation shows that primary ice production plays a key role in generating HIWC regions at temperatures $< -40°C$, shattering of freezing droplets dominates ice particle production in HIWC regions at temperatures between $-15°C$ and $0°C$ during the early stage of convection, and fragmentation during ice–ice collisions dominates at temperatures between $-15°C$ and $0°C$ during the later stage of convection and at temperatures between $-40°C$ and $-20°C$ over the whole convection period. This study confirms the dominant role of SIP processes in the formation of numerous small crystals in HIWC regions.

## 1   Introduction

Homogeneous nucleation of supercooled droplets or heterogeneous nucleation on the surface of ice nucleating particles (INPs) can effectively produce primary ice crystals at temperatures $< -35°C$ (Koop et al., 2000; DeMott et al., 2016). At temperatures $> -35°C$, heterogeneous nucleation on INPs dominates the ice nucleation process and primary ice production. However, many airborne in-situ observations indicate that the observed number concentration of ice crystals often exceeds the concentration of INPs by several orders of magnitude (e.g., Murgatroyd and Garrod, 1960; Hallett and Mossop, 1974; Hobbs and Rangno, 1985; Field et al., 2001; Cantrell and Heymsfield, 2005; Rangno and Hobbs, 2001; Lloyd et al., 2015; Ladino et al., 2017). The concentration of ice particles exceeding the concentration of INPs still persists even though new techniques are applied to mitigate the contamination of ice particle shattering on airborne instruments (Korolev et al., 2013a, b; Korolev and Field, 2015).

Secondary ice production (SIP), also known as "ice multiplication", which produces new ice crystals involving preexisting ice particles, has been recognized as an important mechanism to explain the discrepancy between the concentrations of observed ice particles and INPs (Field et al., 2017; Korolev and Leisner, 2020). Several SIP mechanisms have been described. Korolev and Leisner (2020) summarized laboratory studies of six different SIP mechanisms, namely, (1) the rime-splintering or Hallett–Mossop (H-M) process, (2) ice–ice collision fragmentation, (3) shattering of freezing droplets, (4) fragmentation of sublimating ice particles, (5) ice particle fragmentation due to thermal shock, and (6) activation of INPs in transient supersaturation around freezing drops. Field et al. (2017) discussed the airborne in-situ and radar remote sensing observations, laboratory investigations, and modeling studies of the first four SIP mechanisms in detail. However, the physical basis of these SIP processes remains poorly understood, and quantification of their production rates is not consistent among different studies.

The H-M mechanism, fragmentation of ice–ice collision, and shattering of freezing droplets are the three SIP mechanisms that are often parameterized in numerical models to examine their roles in ice particle production in different types of clouds. The H-M process is the best characterized and has gained extensive attention and evaluation during the last several decades (Field et al., 2017). It results from the collection of liquid drops by ice particles and splintering at a temperature range of about $-3$ to $-8°C$ with a maximum ice particle generation rate at about $-5°C$ (Hallett and Mossop, 1974; Heymsfield and Mossop, 1984). There are two laboratory observations investigating ice–ice collisional fragmentation in different experimental setups and environmental conditions (Vardiman, 1978; Takahashi et al., 1995). Three different parameterization methods have been proposed to represent the fragmentation due to collision of ice particles. In these methods, the number of ice fragments produced per collision is simply fit to the observations of Takahashi et al. (1995) (Sullivan et al., 2017, 2018b), set to a

constant value (Hoarau et al., 2018) or dependent upon the initial kinetic energy and colliding particles' size and rimed fraction (Phillips et al., 2017a, b). The shattering of freezing droplets, the first proposed SIP mechanism, generates small ice splinters following droplet freezing, in which a closed ice shell is formed, freezes inward, and subsequently shatters (Langham and Mason, 1958; Mason and Maybank, 1960; Lauber et al., 2018). Lauber et al. (2018) summarized previous laboratory studies to show that overall fragmentation frequency and the number of ejected splinters per fragmenting droplet increase with increasing droplet diameter. Although the splinter production rates vary with temperature among the laboratory experiments conducted by different research groups, most laboratory studies showed that the freezing droplet fragmentation may be most efficient at temperatures below $-10°C$, outside the temperature range of H-M process (Mason and Maybank, 1960; Brownscombe and Thorndike, 1968; Takahashi and Yamashita, 1969, 1970; Kolomeychuk et al., 1975; Pruppacher and Schlamp, 1975; Takahashi, 1975; Lauber et al., 2018; Keinert et al., 2020).

Several numerical studies investigated the roles of different SIP mechanisms in ice particle production in different types of clouds in different regions (Phillips et al., 2017a, 2018; Sullivan et al., 2018a; Fu et al., 2019; Sotiropoulou et al., 2020, 2021). Phillips et al. (2017a) indicated that the average ice number concentration at temperatures between $\sim 0$ and $-30°C$ increased by one to two orders of magnitude with the inclusion of fragmentation in ice–ice collisions in a cloud-resolving model simulating a multicellular convective storm observed over the U.S. high plains during the Severe Thunderstorm Electrification and Precipitation Study (STEPS). Phillips et al. (2018) used a parcel model to simulate tropical maritime deep convective clouds observed during the Ice in Clouds Experiment-Tropical (ICE-T) field campaign to reveal that fragmentation during raindrop freezing can enhance the number of ice particles initiated at temperatures between 0 and $-20°C$ by one order of magnitude, which dominates the number sources of ice crystals followed by the H-M process. Sullivan et al. (2018a) found that SIP processes contribute to the number concentrations of ice crystals as large as primary ice nucleation, and the H-M process was the most important process in the simulation of a cold frontal rainband observed during the Aerosol Properties, PRocesses And InfluenceS on the Earth's climate (APPRAISE) campaign in the UK. Both Fu et al. (2019) and Sotiropoulou et al. (2020) indicated freezing-drop shattering is insignificant in simulations of Arctic clouds, while Sotiropoulou et al. (2020) found only the combination of H-M and ice–ice collision fragmentation can explain the observed number concentration of ice crystals. Sotiropoulou et al. (2021) suggested that fragmentation during ice–ice collision could account for the high number concentration of ice crystals when the H-M process was weak in the simulation of summer Antarctic mixed-phase clouds. Therefore, although roles of different SIP mechanisms in the production of ice particles differ for different types of clouds, inclusion of these SIP processes in numerical models can indeed explain the discrepancy between the observed number concentrations of ice crystals and INPs to some extent.

The high ice water content (HIWC) phenomenon frequently occurs in tropical oceanic convective clouds, in which there are numerous small ice crystals with median mass diameters (MMDs) of 200–300 $\mu$m, equivalent radar reflectivities ($Z_e$) often < 20 dBZ, and ice water contents (IWCs) often > 1.5 g m$^{-3}$ (Ackerman et al., 2015; Fridlind et al., 2015; Protat et al., 2016; Leroy et al., 2016, 2017; Strapp et al., 2020, 2021). Previous numerical studies using different models and microphysics schemes have indicated that HIWC phenomenon cannot be captured well by numerical models (Franklin et al., 2016; Stanford et al., 2017; Qu et al., 2018; Huang et al., 2021). Huang et al. (2021) evaluated simulations of tropical deep convective clouds

observed on 26 May 2015 during the High Altitude Ice Crystals – High Ice Water Content (HAIC-HIWC) international field campaign using the Weather Research and Forecasting (WRF) model at horizontal grid spacing of 1 km with different bulk microphysics schemes including the Predicted Particle Properties (P3) scheme (Morrison and Milbrandt, 2015; Milbrandt and Morrison, 2016). All of their simulations overestimated the radar reflectivity and underestimated the number concentration of ice particles in HIWC regions compared to the observations. They hypothesized that these biases could be attributed to the poor representation of SIP processes in the microphysics schemes. As a companion paper of Huang et al. (2021), the roles of different SIP mechanisms, namely the H-M mechanism, shattering of freezing droplets, and fragmentation of ice–ice collisions, in the formation of numerous small crystals in HIWC regions are investigated in the current study through a series of sensitivity experiments with the P3 microphysics scheme.

The next section describes the implementation of SIP mechanisms in the P3 scheme and sensitivity experiments conducted in this study. The results of the sensitivity experiments are discussed in Section 3, followed by a summary and conclusions presented in section 4.

## 2   Methodology

### 2.1   Implementation of SIP parameterizations

Milbrandt and Morrison (2016) expanded the P3 scheme to include multiple "free" ice categories, in which particle populations with different sets of bulk properties are allowed to coexist and the detrimental effects of bulk property dilution, where information from particles' different growth paths is lost due to a single set of bulk properties, can be reduced. Their simulation results indicated that at least two ice categories are required to correctly represent the rime splintering process and reduce the bulk property dilution effects. Therefore, the P3 scheme with two ice categories (P3-2ICE) is adopted in this study. The three often parameterized SIP mechanisms in numerical models, namely the H-M mechanism, shattering of freezing droplets, and fragmentation of ice–ice collision, are implemented in the P3-2ICE scheme. There are other SIP mechanisms reviewed by Korolev and Leisner (2020) that are not considered in the simulations presented here, such as fragmentation of sublimating ice particles, ice particle fragmentation due to thermal shock, and activation of INPs in transient supersaturation around freezing drops. However, to the best of our knowledge, currently there is only one recent attempt to parameterize fragmentation of sublimating ice particles (Deshmukh et al., 2021) and there are no parameterizations for the other two SIP mechanisms, so it is difficult to implement them in simulations. It also should be noted that different SIP mechanisms operate efficiently in different conditions, which are functions of environmental temperature, existence of drops, and ice particle sizes, etc. Further, there can be competition between different SIP mechanisms operating at similar conditions, such as two mechanisms requiring the involvement of raindrops. Therefore, adding other SIP mechanisms would not necessarily lead to higher ice number concentration.

The H-M mechanism was parameterized in the original P3 scheme, and it is switched on when multiple ice categories are used. The parameterization of the H-M mechanism follows Cotton et al. (1986) and is based on the laboratory study of Hallett and Mossop (1974), in which ∼350 splinters were produced per 1 mg of accreted liquid at $-5°C$. In the parameterization,

the maximum splinter production rate due to the H-M mechanism (350 per 1 mg of the accreted water) is assumed at an ambient temperature of $-5°C$ and linearly decreases to zero at $-3$ and $-8°C$. Atlas et al. (2020) used the double-moment Morrison scheme (Morrison et al., 2005) to simulate the boundary layer clouds over the summertime Southern Ocean and recommended that removing all of the thresholds associated with the mixing ratios of liquid and frozen hydrometeors in the H-M parameterization to activate the H-M process within the H-M temperature range. In the original H-M parameterization of P3-2ICE, the H-M process is activated only when ice mean-mass diameter $> 4$ mm, which is rarely observed (Huang et al., 2021). Thus, the threshold of ice mean-mass diameter in the H-M parameterization is removed in this study.

The parameterization of freezing-droplet shattering is implemented using the numerical formulation proposed by Phillips et al. (2018), which combined observations from previous laboratory studies and considered the physics of collisions. Two modes of the scheme, fragmentation during heterogeneous drop freezing (mode 1) and accretion of raindrops (mode 2), are considered in this study. The number of fragments per frozen drop in mode 1 is dependent on raindrop size and freezing temperature, and it is dependent on collision kinetic energy in addition to raindrop size and freezing temperature in mode 2 (Phillips et al., 2018). The raindrop-freezing fragmentation scheme is implemented in P3 by adopting a bin-emulating approach, in which bulk particle size distributions are discretized into bins for the calculations of microphysical process rates (Saleeby and Cotton, 2008; Morrison, 2012). A more detailed description of this parameterization is found in Phillips et al. (2018).

The ice-ice collection or aggregation (collision and coalescence) process was considered in the original P3 scheme, but fragmentation during ice–ice collision was not. The physically-based parameterization of ice multiplication by breakup during ice–ice collision proposed by Phillips et al. (2017b) is adopted and implemented in the P3 scheme using a bin-emulating approach. The scheme is based on an energy conservation principle, in which the number of new fragments per collision is dependent on the cloud species (i.e., hail, graupel, snow or crystals whether dendritic or spatial planar), collision kinetic energy, temperature, and colliding particles' size and rimed fraction. Parameters in the scheme are estimated based on previous laboratory and field experiments (Vardiman, 1978; Takahashi et al., 1995), with more details found in Phillips et al. (2017b). The collection (aggregation) efficiency ($E_{agg}$) between ice particles follows the laboratory study of Connolly et al. (2012), in which $E_{agg}$ are 0.09, 0.21, 0.6, 0.1, 0.08, 0.02 at temperatures of $-5, -10, -15, -20, -25, -30°C$, respectively. When temperature $> -5°C$ and $< -30°C$, $E_{agg}$ is set to 0.09 and 0.02 respectively, and otherwise $E_{agg}$ is linearly interpolated between temperatures. As with most bulk schemes, the collision efficiency between ice particles is assumed to be 1, implying $E_{agg}$ is equal to the coalescence efficiency ($E_{coal}$). Therefore, ice–ice collision breakup efficiency is equal to $1 - E_{coal}$. Field et al. (2006) indicated that a constant aggregation efficiency of 0.09 ($E_{agg} = 0.09$) produced good agreement with aircraft observations, and Morrison and Grabowski (2010) assumed $E_{agg} = 0.1$ in their study. The main results and conclusions do not change in sensitivity experiments using a constant $E_{agg} = 0.1$. Therefore, only results using $E_{agg}$ following Connolly et al. (2012) are shown in this paper.

In this study, current commonly accepted microphysical parameterizations are used. However, there are uncertainties in the parameterization of both primary ice production and SIP mechanisms (Korolev and Leisner, 2020). In fact, uncertainties in the parameterization of primary ice production also transfer to uncertainties in SIP processes. Therefore, more theoretical studies,

field campaigns including remote-sensing and in-situ observations, and laboratory studies should be conducted to constrain parameterizations of both primary ice production and SIP mechanisms in the future (Morrison et al., 2020).

## 2.2 Numerical experiments

Idealized experiments that consume less computing resources are conducted first, and then the optimal configurations from these idealized studies are used to rerun a real-case experiment to examine whether changes to the default P3 scheme can improve the simulation of HIWC phenomenon.

### 2.2.1 Idealized simulation

The WRF version 4.1.3, as used by Huang et al. (2021), is employed in a three-dimensional quasi-idealized framework to simulate the tropical oceanic convection observed on 26 May 2015. The input sounding used for the initial horizontally-uniform thermodynamic environment is from a radiosonde released at Cayenne at 00:00 UTC 26 May 2015 (Fig. 1a). The sounding has a deep moist absolutely unstable layer and mainly easterly (westerly) winds below (above) 350 hPa. The surface-based convective available potential energy of the sounding is 2378 J kg$^{-1}$.

The model domain is $200 \times 100$ km$^2$ with horizontal grid spacings between 125 and 1000 m and the model top is 18 km with 71 vertical levels. The model time step is 1 s. Three dimensional subgrid-scale mixing is calculated using a 1.5-order turbulent kinetic energy scheme (Skamarock et al., 2019) instead of a planetary boundary layer (PBL) scheme. An ocean surface is assumed and the surface moisture and sensible/latent heat fluxes are estimated using the MM5 similarity surface layer scheme (Jiménez et al., 2012). The other physical parameterization schemes, including longwave and shortwave radiation scheme, land-surface scheme and cumulus parameterization scheme, are not activated in the idealized simulations. The P3 two-ice microphysics scheme is adopted and the detailed setups are described in Section 2.2.2.

Updraft nudging (Naylor and Gilmore, 2012) is adopted to initiate convection within the horizontally-uniform thermodynamic environment. The updraft ($w_t$) within a spheroid with 10-km horizontal radius ($x_r = y_r = 10$ km) and 1.5-km vertical radius ($z_r = 1.5$ km) centered at $z_c = 1.5$ km is determined by

$$w_t = w_{t-1} + \Delta t \times \alpha \times \gamma \times \max\left(w_{\mathrm{mag}} - w_{t-1}, 0\right), \tag{1}$$

with

$$w_{\mathrm{mag}} = \begin{cases} w_{\max} \cos^2\left(\frac{\pi}{2}\beta\right), & \text{if} \quad 0 \le \beta \le 1, \\ 0, & \text{if} \quad \beta > 1, \end{cases} \tag{2}$$

where $\Delta t$ is the small model time step, $\alpha = 0.5$ s$^{-1}$, $w_{\max} = 10$ m s$^{-1}$, $\beta = \sqrt{\left(\frac{x-x_c}{x_r}\right)^2 + \left(\frac{y-y_c}{y_r}\right)^2 + \left(\frac{z-z_c}{z_r}\right)^2}$, and $x_c$ and $y_c$ are the horizontal locations at the domain center. Updraft nudging starts at $t = 0$ and lasts 20 min. The coefficient $\gamma$ is a function of time $t$ in the unit of min, and $\gamma$ is defined by

$$\gamma = \begin{cases} 1, & \text{if} \quad t < 15 \text{ min}, \\ (20-t)/5, & \text{if} \quad 15 \le t \le 20 \text{ min}. \end{cases} \tag{3}$$

## 2.2.2 Sensitivity experiments

The smallest horizontal grid spacing used in the simulations of Huang et al. (2021) was 1 km, which is not cloud-resolving $O(100\,\text{m})$. At a grid spacing of $O(1\,\text{km})$, horizontal entrainment and mixing is under-represented (Bryan and Morrison, 2012; Lebo and Morrison, 2015), which influences the liquid water content (LWC) available for riming growth. Lebo and Morrison (2015) found overall storm characteristics had limited sensitivity when horizontal grid spacing was decreased below 250 m in their simulated squall lines. Jeevanjee (2017) indicated that horizontal resolutions of $O(100\,\text{m})$ can be required for convergence of convective vertical velocities. Therefore, three sensitivity experiments using different horizontal grid spacings (i.e., 1000, 250 and 125 m) are conducted to investigate whether higher resolution simulations can reduce the simulated biases in ice number concentration shown in Huang et al. (2021). In these three sensitivity experiments, referred to as NoSIP1kmAC, NoSIP250mAC and NoSIP150mAC, respectively, all SIP processes are turned off and the default constant aerosol number mixing ratio (i.e., the ratio of the aerosol number concentration and air density $= 300 \times 10^6\,\text{kg}^{-1}$) in the original P3 scheme is used.

Huang et al. (2021) indicated that the simulation with the P3 scheme overestimates the LWC at $-10^\circ\text{C}$, which enhances the collection of liquid water by ice particles and subsequently increases the mass/size of ice particles but not ice particle number. Ladino et al. (2017) showed the aerosol concentration decreases with increasing of height, and that aerosol concentrations are about 360 and 50 $\text{cm}^{-3}$ within the boundary layer and the free troposphere respectively through the vertical profile of aerosols averaged over the entire Ultra-High Sensitivity Aerosol Spectrometer (UHSAS) dataset during the Cayenne campaign (Fig. 2 in Ladino et al., 2017). Thus, the overestimate of the simulated LWC may be associated with the relatively larger aerosol number mixing ratio above the boundary layer used in the original P3 scheme. A sensitivity experiment using a more realistic profile of aerosol number mixing ratio based on the in-situ observations (Ladino et al., 2017), instead of the constant value in the original P3 scheme, is performed to explore whether it can reduce the simulated biases in LWC and ice number concentration at $-10^\circ\text{C}$. In this experiment, referred to as NoSIP250m hereafter, a horizontal grid spacing of 250 m is used and the vertical profile of aerosol number mixing ratio is shown in Fig. 1b.

Another four sensitivity experiments on SIP processes are performed with a horizontal grid spacing of 250 m and the more realistic vertical profile of aerosol number mixing ratio. They are experiments with only the H-M process on, only the raindrop freezing breakup process on, only the ice-ice collision breakup process on, and all SIP processes on, referred to as HM250m, RFZB250m, IICB250m, and SIPs250m hereafter, respectively. These experiments are conducted to examine the processes leading to the production of large numbers of small ice crystals. All sensitivity experiments in this study are summarized in Table 1. A horizontal grid spacing of 250 m and the more realistic vertical profile of aerosol number mixing ratio are chosen for the sensitivity experiments including SIP processes, because results reveal the model resolution and aerosol profile are not the main source of model biases in simulating HIWCs (discussed in detail in Section 3.2), and because a simulation using 125-m grid spacing consumes much more computing resources than a simulation using 250-m grid spacing.

Figure 2 shows the evolution of composite reflectivity in SIPs250m. After the convection initiation, it develops into deep convection and gradually reaches a mature stage at $t = 60$ min (Figs. 2a–c). The convection further develops, broad anvil clouds form, and finally the convection begins to weaken at $t = \sim110$ min (Figs. 2d–f).

## 3 Results

### 3.1 Observations

Figure 3 shows scatter plots of observed ice number concentration (Ni) for maximum particle dimensions ($D_{\mathrm{max}}$) between 0.1 and 12.845 mm (Huang et al., 2021) divided by IWC (denoted as Ni/IWC hereafter) as a function of vertical velocity in regions with IWC $> 1$ g m$^{-3}$ from all flights during the Cayenne field campaign at temperatures of $-10$, $-30$, and $-45°$C. The observed temperature ranges of samples at the three levels are $-12.9$ to $-7.3°$C, $-33.0$ to $-27.3°$C, $-45.4$ to $-42.4°$C, respectively. These flights mainly sampled the mature stage of convection, and the observations are analyzed using sampling windows of 5 s, corresponding to a grid spacing of $\sim900$ m with a typical aircraft horizontal speed of 180 m s$^{-1}$ (Hu et al., 2021). Cloud segments with the presence of liquid water were identified from voltage changes of the Rosemount Icing Detector and from the total concentration measured by the Cloud Droplet Probe version 2 (CDP-2), and not considered in this analysis. Composite particle size distributions were derived from the Two Dimensional Stereo Imaging Probe (2D-S) and the Precipitation Imaging Probe (PIP) for the particles with $D_{\mathrm{max}}$ between 0.01 and 12.845 mm. The observed Ni only considers contributions from ice crystals with $D_{\mathrm{max}} > 0.05$ mm due to the potential of shattered artifacts and small and poorly defined depth of field for small particles (Huang et al., 2021; Hu et al., 2021). There is considerable uncertainty in estimating concentrations of ice crystals with $D_{\mathrm{max}} < 0.2$ mm from current probes and processing algorithms (McFarquhar et al., 2017; O'shea et al., 2021). To examine the sensitivity of findings to ice crystal concentrations in small sizes, sensitivity tests using different lower limits of $D_{\mathrm{max}}$ (i.e., 0.05, 0.1, and 0.2 mm) were conducted. The qualitative findings are consistent among these sensitivity tests (Figs. S1–S6), so only results using the lower limit of $D_{\mathrm{max}} = 0.1$ mm are discussed here. More details on the processing and uncertainty of observations can be found in Huang et al. (2021) and Hu et al. (2021).

At the $-10°$C level, Ni/IWC covers three orders of magnitude between $10^3$ and $10^6$ g$^{-1}$ and $\sim53.9$ % and $\sim45.8$ % of samples are between $10^4$ and $10^5$ g$^{-1}$ and between $10^5$ and $10^6$ g$^{-1}$, respectively. With an increase of either upward or downward vertical velocity, Ni/IWC increases, passing the $t$-test for $p < 0.05$ (Fig. 3a). At the $-30°$C level, Ni/IWC covers two orders of magnitude between $10^4$ and $10^6$ g$^{-1}$, and $\sim85.4$ % of samples are between $10^5$ and $10^6$ g$^{-1}$ (Fig. 3b). With an increase of either upward or downward vertical velocity, Ni/IWC increases at $-30°$C (Fig. 3b, passing the $t$-test for $p < 0.05$), but the slope is less than that at $-10°$C (Figs. 3a and b). At the $-45°$C level, $\sim98.4$ % of Ni/IWC samples are between $10^5$ and $10^6$ g$^{-1}$ (Fig. 3c). Ni/IWC does not appear to increase with vertical velocity at $-45°$C (Fig. 3c, not passing the $t$-test for $p < 0.05$) in contrast to results at temperatures of $-10$ and $-30°$C.

## 3.2 Sensitivity on horizontal resolution and aerosol profile

Figure 4 shows scatter plots of simulated Ni/IWC for 0.1 mm $< D_{\max} <$ 12.845 mm as a function of vertical velocity in regions with IWC $> 1$ g m$^{-3}$ linearly interpolated to the temperatures of $-10$, $-30$, and $-45°$C at $t = 60$ min in experiments of NoSIP1kmAC, NoSIP250mAC, NoSIP125mAC, and NoSIP250m, respectively. The convection at $t = 60$ min is at the mature stage, which is consistent with the observations (Hu et al., 2021). Similar to Huang et al. (2021), ice number distribution function is attained from the model to re-calculated the related variables for the same range and same bin size of $D_{\max}$ as the observed. Here, the simulations with horizontal grid spacing $< 1$ km have been coarsened to 1 km, similar to the grid spacing of observations, for comparison by spatially averaging with a window size of 1 km $\times$ 1 km. It should be noted that the coarsened results are similar to those at the original grids (not shown).

At the $-10°$C level, the simulations excluding SIP processes produce Ni/IWC mainly covering two orders of magnitude between $10^2$ and $10^4$ g$^{-1}$ (Figs. 4a1–d1), which is about two orders of magnitude less than observed (Fig. 3a). With an increase of upward vertical velocity or decrease of downward vertical velocity, Ni/IWC has a decreasing trend (Figs. 4a1–d1), which is also different from the observations (Fig. 3a). In addition, the radar reflectivities in these simulations are mainly greater than 35 dBZ at $-10°$C (Figs. 4a1–d1), which is overestimated compared to the observations in HIWC regions where 95 % of the cumulative observed reflectivities are $< 30$ dBZ (Huang et al., 2021). Therefore, at $-10°$C the experiments without SIP processes using any horizontal grid spacing (Figs. 4a1–c1) or any aerosol profile (Figs. 4b1 and d1) cannot produce HIWC regions consistent with observations. There are no obvious differences among these simulations, although the number concentration of cloud droplets is reduced by $\sim$74.5 % in the NoSIP250m experiment using the aerosol profile based on UHSAS observations (not shown), which is closer to the observed.

At the $-30°$C level, Ni/IWC in the simulations without SIP processes are mainly distributed between $10^4$ and $10^5$ g$^{-1}$ with $\sim$11 % $> 10^5$ g$^{-1}$ in NoSIP1kmAC and no samples with Ni/IWC $> 10^5$ g$^{-1}$ in the other simulations (Figs. 4a2–d2), which is about an order of magnitude less than the observations (Fig. 3b). About 59 %, 84 %, 82 % and 83 % of radar reflectivities at $-30°$C are greater than 30 dBZ in NoSIP1kmAC, NoSIP250mAC, NoSIP125mAC, and NoSIP250m, respectively (Figs. 4a2–d2). Therefore, the simulations without SIP processes cannot produce HIWC regions at $-30°$C.

At the $-45°$C level, Ni/IWC values in the simulations without SIP processes are mainly distributed between $10^5$ and $10^6$ g$^{-1}$ (Figs. 4a3–d3), which is consistent with the observations (Fig. 3c). However, the magnitude of the simulated Ni/IWC is mainly around $10^6$ g$^{-1}$, which is greater than observed. About 80 %, 94 %, 100 %, 88 % of radar reflectivities at $-45°$C are less than 10 dBZ in NoSIP1kmAC, NoSIP250mAC, NoSIP125mAC, and NoSIP250m, respectively (Figs. 4a3–d3). These results indicate that the simulations without SIP processes can produce HIWC regions at $-45°$C, however, these simulations obviously overestimate the Ni/IWC at this level.

Overall, the simulations without SIP processes underestimate Ni/IWC and overestimate radar reflectivity at temperatures of $-10$ and $-30°$C, that is, they cannot produce HIWC regions at these temperature levels. These simulations can produce HIWC regions at $-45°$C, but they overestimate Ni/IWC at this level. These results are not sensitive to model horizontal grid spacing

or aerosol profile. The biases in the simulations without SIP processes are seen clearly in Fig. A1 in which simulations are overlaid with observations.

Previous studies (e.g., D'Alessandro et al., 2017; Diao et al., 2017) showed that the ice supersaturation threshold in the ice nucleation parameterization of Cooper (1986) used in common microphysics schemes (e.g., Morrison et al., 2005; Morrison and Milbrandt, 2015) is too low, which can affect the distribution of ice water content and ice number concentration substantially. To examine the impact of varying this threshold, a sensitivity study changing the ice supersaturation threshold from 5 % to 25 % in the ice nucleation parameterization of the P3 scheme was conducted. The simulation is the same as NoSIP250m but the ice supersaturation threshold of 25 % used in the ice nucleation parameterization (referred to as NoSIP250mIS25). Figure 5 shows scatter plots of simulated Ni/IWC for $0.1$ mm $< D_{\max} < 12.845$ mm as a function of vertical velocity in regions with IWC $> 1$ g m$^{-3}$ linearly interpolated to the temperatures of $-10$, $-30$, and $-45°$C at $t = 60$ min in NoSIP250mIS25 overlaid with observations in Fig. 3. From Fig. 5, the results in NoSIP250mIS25 are very similar to those in NoSIP250m (Figs. 4d1–d3), in terms of orders of magnitude of Ni/IWC and intensity of radar reflectivity at the three temperature levels. It indicates that changing ice supersaturation threshold in the ice nucleation parameterization does not influence the conclusions attained in this study.

## 3.3 Sensitivity on including SIP processes

Figure 6 shows scatter plots of simulated Ni/IWC for $0.1$ mm $< D_{\max} < 12.845$ mm as a function of vertical velocity in regions with IWC $> 1$ g m$^{-3}$ linearly interpolated to temperatures of $-10$, $-30$, and $-45°$C at $t = 60$ min in experiments HM250m, RFZB250m, IICB250m, and SIPs250m, respectively. Similarly, the simulations have been coarsened to 1-km grid spacing to compare with the observations, and the conclusions are not influenced by the coarsened process.

At the $-10°$C level, Ni/IWC values in the simulations with at least one SIP process increase significantly (Figs. 6a1–d1) compared to the simulations without SIP processes (Figs. 4a1–d1). However, in HM250m $\sim$99 % of Ni/IWC values are less than $10^5$ g$^{-1}$, and it does not have an obviously increasing trend with an increase of downward vertical velocity (Fig. 6a1, not passing the $t$-test for $p < 0.05$), which is inconsistent with the observations (Fig. 3a). About 72 % of radar reflectivities in HM250m are greater than 30 dBZ, which are overestimated compared to observed in HIWC regions at $-10°$C, where 95 % of the cumulative observed reflectivities are $< 30$ dBZ (Huang et al., 2021). In RFZB250m, Ni/IWC covers three orders of magnitude between $10^3$ and $10^6$ g$^{-1}$ (Fig. 6b1), which is consistent with the observations (Fig. 3a). However, RFZB250m does not produce the observed relationship between Ni/IWC and downward vertical velocity, which is similar to HM250m. About 63 % of radar reflectivities in RFZB250m are less than 30 dBZ (Fig. 6b1). Thus, RFZB250m can produce HIWC regions at $-10°$C to a certain extent. In IICB250m (Fig. 6c1), Ni/IWC covers four orders of magnitude between $10^2$ and $10^6$ g$^{-1}$, which is underestimated compared to observations especially at larger vertical velocities. Meanwhile, IICB250m fails to capture the observed relationship between Ni/IWC and upward vertical velocity. However, IICB250m produces $\sim$30 % of samples with HIWC characteristics, that is, high Ni/IWC $> 10^5$ g$^{-1}$ and radar reflectivities $< 20$ dBZ (Fig. 6c1). In SIPs250m that includes all three SIP mechanisms (Fig. 6d1), Ni/IWC covers the same range as observed (i.e., between $10^3$ and $10^6$ g$^{-1}$), and the increase of Ni/IWC with greater upward or downward vertical velocity is also captured well. Around 96 % of

radar reflectivities in SIPs250m are less than 30 dBZ, indicating that SIPs250m produces HIWC regions at $-10°$C remarkably consistent with the observations.

At the $-30°$C level, Ni/IWC values in the simulations with at least one SIP process increase up to $\sim10^6$ g$^{-1}$ (Figs. 6a2–d2), which is the same as the observations (Fig. 3b). However, $\sim83.2$ % and $\sim57.4$ % of Ni/IWC values in HM250m and RFZB250m are less than $10^5$ g$^{-1}$, respectively (Figs. 6a2 and b2), which differs from the observations whose samples are mainly ($\sim85.4$ %) distributed between $10^5$ and $10^6$ g$^{-1}$ (Fig. 3b). Although $\sim92.7$ % of Ni/IWC values in IICB250m are distributed between $10^5$ and $10^6$ g$^{-1}$, there are $\sim5.6$ % of samples with Ni/IWC $< 10^5$ g$^{-1}$ and stronger radar reflectivities $> 30$ dBZ. This was not observed during the Cayenne field campaign (Fig. 3b). Even so, the HM250m, RFZB250m, and IICB250m simulations produce $\sim16.8$ %, $\sim38.6$ %, and $\sim90.3$ % of samples with HIWC characteristics, that is, with Ni/IWC $> 10^5$ g$^{-1}$ and radar reflectivity $< 20$ dBZ (Figs. 6a2–c2). SIPs250m produces $\sim89.6$ % of samples with Ni/IWC between $10^5$ and $10^6$ g$^{-1}$ and radar reflectivities $< 20$ dBZ (Fig. 6d2), which is consistent with observations of HIWC regions ($\sim85.4$ %) at $-30°$C.

At the $-45°$C level, compared to the observations (Fig. 3c), HM250m and RFZB250m produce a broader range of Ni/IWC, especially HM250m covering three orders of magnitude between $10^3$ and $10^6$ g$^{-1}$ (Figs. 6a3 and b3). IICB250m overestimates Ni/IWC, with $\sim5.8$ % of Ni/IWC values $< 6 \times 10^5$ g$^{-1}$ while $\sim97.9$ % of observed Ni/IWC values $< 6 \times 10^5$ g$^{-1}$ (Figs. 6c3 and 3c). Regardless of the bias in Ni/IWC in HM250m, RFZB250m, and IICB250m, these simulations produce $\sim86.9$ %, $\sim92.2$ %, and 100 % of samples with radar reflectivity $< 20$ dBZ at $-45°$C (Figs. 6a3–c3). SIPs250m simulates $\sim98.4$ % of Ni/IWC values between $10^5$ and $10^6$ g$^{-1}$ and radar reflectivities less than 20 dBZ (Fig. 6d3), similar to observed. This indicates SIPs250m successfully produces HIWC regions at $-45°$C.

To further examine the role of SIP mechanisms in different locations of the convective storm, we analyze a vertical cross section through the convective core (based on the maximum composite reflectivity) of microphysical process rates relevant to ice particle production including: the H-M mechanism, shattering of freezing droplets, fragmentation of ice–ice collisions, and other microphysical processes (i.e., primary ice nucleation, homogeneous and heterogeneous freezing of cloud droplets and rain). Results from SIPs250m are shown in Fig. 7 for regions with IWC $> 1$ g m$^{-3}$. The H-M process (mainly at $-5°$C) and shattering of freezing droplets (mainly at temperatures between $-5$ and $-20°$C) dominate ice particle production ($> 58$ %) in the strong updraft core regions where there is plentiful LWC. Fragmentation during ice–ice collisions is dominant ($\sim100$ %) in the other HIWC regions (Fig. 7). In general, total ice particle production rates are about 4 times larger in the strong updraft regions ($w >10$ m s$^{-1}$) than those in other HIWC regions. The importance of freezing fragmentation enhanced by updrafts is consistent with an observational study on mixed-phase clouds at temperatures $> -10°$C in the Arctic (Luke et al., 2021).

Overall, the simulations including only one of the three SIP mechanisms – the H-M process, shattering of freezing raindrops, or fragmentation during ice-ice collisions – cannot fully explain the observed HIWC characteristics at temperatures of $-10$, $-30$, and $-45°$C. Only the simulation including all three SIP mechanisms, (i.e., SIPs250m) can successfully capture the observed HIWC regions at the three temperature levels. The good agreement between SIPs250m and observations can be seen clearly in Fig. A2 in which simulations are overlaid with observations.

Because SIP processes need to be triggered by preexisting ice and the ice-ice collision process is strongly dependent on Ni, the relative contribution of SIP processes to ice particle production should be different at different stages of convection. Figure 8 shows the time evolution of the microphysical process rates relevant for ice particle production including the H-M mechanism, shattering of freezing droplets, fragmentation of ice–ice collisions, and other microphysical processes (i.e., primary ice nucleation, homogeneous and heterogeneous freezing of cloud droplets and rain) in regions with IWC $> 1$ g m$^{-3}$ at

different temperatures in SIPs250m. It indicates that the roles of SIP processes in ice particle production in HIWC regions vary during the evolution of convection. At the early stage of convection (t $< 40$ min), primary ice production (mainly homogeneous freezing of cloud droplets) dominates ice particle production ($> 50$ % of total ice particle production rate) at temperatures less than $-40°$C, fragmentation of ice–ice collisions is dominant ($> 50$ % of total ice particle production rate) at temperatures between $-40$ and $-20°$C and at $0°$C, and shattering of freezing droplets plays the key role ($> 50$ % of total ice particle

production rate) at temperatures between $-15$ and $-5°$C (Fig. 8). With the development of convection, Ni increases, and the fragmentation of ice–ice collisions becomes dominant ($> 50$ % and maximum close to 100 % of total ice particle production rate) at temperatures less than $0°$C. The H-M process also plays a role ($\sim5$ % of total ice particle production rate) in the ice particle production around $-5°$C. Therefore, primary ice production is dominant in HIWC regions at the very early stage of convection at temperatures less than $-40°$C, shattering of freezing droplets dominates ice particle production in HIWC regions

at temperatures between $-15°$C and $0°$C during the early stage of convection, and fragmentation during ice–ice collisions is dominant at temperatures between $-15°$C and $0°$C during the later stage of convection and at temperatures between $-40°$C and $-20°$C over the whole convection period.

### 3.4   Improvement in real-case simulation

To examine whether the new P3 two-ice category configuration including all three SIP mechanisms can improve the simulation

of HIWC regions for a real-case study, the experiment P3-2ICE of Huang et al. (2021) using the original P3 two-ice category configuration (referred to as P3-2ICE_ORG hereafter) to simulate the tropical oceanic convective system observed on 26 May 2015 during the HAIC-HIWC field campaign based out of Cayenne, French Guiana, is rerun using the new P3 two-ice category configuration including SIP processes. The new experiment is referred to as P3-2ICE_SIP hereafter. The storm coverage and evolution in P3-2ICE_SIP are consistent with those in P3-2ICE_ORG, resembling the observations (not shown). Figure 9

shows scatter plots of observed and simulated Ni/IWC as a function of vertical velocity in regions with IWC $> 1$ g m$^{-3}$ at temperatures of $-10$, $-30$, and $-45°$C. The temperature ranges of observed samples at the three levels are $-12.6$ to $-7.9°$C, $-30.4$ to $-29.7°$C, $-44.7$ to $-43.6°$C, respectively. The simulations at the three temperature levels are interpolated from the model outputs. The simulations are from the 1-km domain of P3-2ICE_ORG and P3-2ICE_SIP at 10:45 UTC 26 May 2015, when the storm was at the the mature stage and observed by two flights, SAFIRE Falcon 20 and NRC Convair 580 (shown in

Fig. 1 of Huang et al. (2021)), during the Cayenne field campaign.

From Fig. 9, P3-2ICE_ORG underestimates Ni/IWC by about two orders of magnitude at $-10°$C and one order of magnitude at $-30°$C (Figs. 9a1 and a2). Although Ni/IWC at $-45°$C in P3-2ICE_ORG covers the observed range between $10^5$ and $10^6$ g$^{-1}$, it covers three orders of magnitude between $10^3$ and $10^6$ g$^{-1}$ with $\sim75.5$ % of Ni/IWC values $< 10^5$ g$^{-1}$ (Fig. 9a3). From

the observed radar reflectivity shown in Fig. 7e of Huang et al. (2021), 95 % of the cumulative observed reflectivities are less than 30 dBZ at $-10^{\circ}$C, less than 20 dBZ at $-30^{\circ}$C, and less than 15 dBZ at $-45^{\circ}$C. However, $\sim$16.4 %, $\sim$0.4 %, and $\sim$4.6 % of the simulated radar reflectivities in P3-2ICE_ORG are less than 30 dBZ at $-10^{\circ}$C, less than 20 dBZ at $-30^{\circ}$C, and less than 15 dBZ at $-45^{\circ}$C, respectively (Figs. 9a1–a3), which are underestimated compared to the observations. In P3-2ICE_SIP, the simulated samples cover all the observed samples at $-10$ and $-30^{\circ}$C (Figs. 9b1–b2). Although the simulated samples at $-45^{\circ}$C in P3-2ICE_SIP do not cover all the observed samples, $\sim$94.9 % and 100 % of Ni/IWC values are distributed between $10^5$ and $10^6$ g$^{-1}$ in P3-2ICE_SIP and observations, respectively (Fig. 9b3). There are $\sim$85.4 %, $\sim$93.0 %, and $\sim$99.1 % of the simulated radar reflectivities in P3-2ICE_SIP $<$ 30 dBZ at $-10^{\circ}$C, $<$ 20 dBZ at $-30^{\circ}$C, and $<$ 15 dBZ at $-45^{\circ}$C, respectively, (Figs. 9b1–b3), which is very consistent with the observed. More small ice particles generated in the early or lower level cloud through SIP processes also can increase the small ice crystals at upper cloud through vertical advection. These results are also consistent with those in the quasi-idealized simulation SIPs250m (Fig. 6d1–d3). Therefore, the real-case simulation using the new P3 two-ice category configuration including all three SIP mechanisms can successfully reproduce the HIWC regions of the observed tropical oceanic convective system. It also confirms the dominant role of SIP processes in HIWC regions with high concentration of small ice crystals.

## 4 Summary and conclusions

A previous study (Huang et al., 2021) used the WRF model at 1-km horizontal grid spacing with four different bulk microphysics schemes to simulate tropical deep convective clouds observed during the HAIC-HIWC field campaign. The simulations overestimated the intensity and spatial extent of radar reflectivity above the melting layer and failed to reproduce the observed high concentrations of small ice crystals in HIWC regions, in which there are numerous small ice crystals with MMDs of 200–300 $\mu$m, $Z_e$ often $<$ 20 dBZ, and IWCs often $>$ 1.5 g m$^{-3}$. To explore formation mechanisms for HIWC regions and biases in the WRF simulations, a series of quasi-idealized sensitivity experiments on the model resolution, aerosol profile, and SIP processes are conducted based on an observed sounding from a radiosonde released at Cayenne during the HAIC-HIWC field campaign. The P3 two-ice category configuration, which has two "free" ice categories to represent all ice-phase hydrometeors, is used. The main results are summarized as follows:

(1) By comparing simulations to observations, regardless of the horizontal grid spacing (1 km, 250 m and 125 m) or aerosol profile used (default constant profile in original P3 scheme or aerosol profile based on UHSAS measurements from HAIC-HIWC field campaign), without SIP processes the model produces total ice number concentrations about two orders of magnitude less than observed at $-10^{\circ}$C and about an order of magnitude less than observed at $-30^{\circ}$C. These simulations also overestimate the radar reflectivity at $-10$ and $-30^{\circ}$C. Although the simulations can produce HIWC regions at $-45^{\circ}$C, they overestimate the ice number concentration compared to observations.

(2) Three simulations turning on one of three SIP mechanisms separately (i.e., the Hallett-Mossop mechanism, fragmentation during ice–ice collisions, and shattering of freezing droplets) can produce higher ice number concentrations at $-10$ and $-30^{\circ}$C,

but they do not fully replicate observations of HIWCs, with the results of the simulation with shattering of freezing droplets most closely resembling the observations.

(3) The simulation including all three SIP processes successfully produces HIWC regions at all temperature levels in terms of ice number concentration and radar reflectivity. Based on a vertical cross section of ice particle production rates through the mature convection, the H-M mechanism (mainly at $-5°$C) and shattering of freezing droplets (mainly at temperature between $-5$ and $-20°$C) dominate ice particle production in strong updraft core regions where there is plentiful LWC, and fragmentation of ice–ice collisions is dominant in the other HIWC regions. Time evolution of the relative contributions of ice crystal sources at different temperature levels indicates that primary ice production plays a role in HIWC regions at the very early stage of convection at temperatures less than $-40°$C, shattering of freezing droplets dominates ice particle production in HIWC regions at temperatures between $-15°$C and $0°$C during the early stage of convection, and fragmentation during ice–ice collisions is dominant at temperatures between $-15°$C and $0°$C during the later stage of convection and at temperatures between $-40°$C and $-20°$C over the whole convection period.

(4) The new P3 two-ice category configuration including all three SIP mechanisms is used for a real-case simulation of the tropical oceanic convective system observed on 26 May 2015 during the HAIC-HIWC field campaign, which was also simulated by Huang et al. (2021) using the original P3 two-ice category configuration. The results indicate that the new P3 two-ice category configuration can reproduce the HIWC regions at all temperature levels in terms of ice number concentration and radar reflectivity, which were not replicated using the original P3 two-ice category configuration.

In conclusion, the model resolution and aerosol profile are not the main source of model biases in simulating HIWCs in tropical deep convective clouds, while SIP processes dominate the high concentrations of small ice crystals in HIWC regions. It should be noted that there might exist uncertainties in the parameterization of SIP mechanisms used in this study. For example, the high ice production rate due to the fragmentation during ice-ice collisions is highly uncertain, and its high production rate in anvil cloud regions between $-40°$C and $-50°$C (Fig. 7) is rarely seen in observations. However, these uncertainties do not influence the main conclusions due to the orders of magnitude differences in ice number concentrations between the experiments with and without SIP mechanisms. This study enhances understanding of the processes leading to formation of the numerous small crystals in HIWC regions in which enhanced production of secondary ice is one of the necessary conditions. In addition, a recent study (Zhao and Liu, 2021) suggested that global climate models including SIP processes can reduce biases in the global annual average liquid/ice water paths and change the global annual average net cloud radiative forcing. Therefore, more theoretical studies, field campaigns including remote-sensing and in-situ observations and laboratory studies should be conducted to constrain parameterization of SIP mechanisms used in numerical weather and climate models further.

*Code and data availability.* The WRF code is available at https://github.com/wrf-model/WRF (last access: 4 January 2020). Observation data are available at https://data.eol.ucar.edu/master_lists/generated/haic-hiwc_2015 (last access: 26 May 2020). ERA5 reanalysis data are available at https://doi.org/10.5065/BH6N-5N20 (European Centre for Medium-Range Weather Forecasts, 2019).

## Appendix A

Figures A1 and A2 as Figs. 4 and 6 respectively but overlaid with observations in Fig. 3.

*Author contributions.*  YoH, WW, and GMM designed the study. YoH implemented the parameterization of secondary ice production for the P3 scheme, conducted the experiments and analyses with support from WW, GMM, MX, HM, JM, and AVK. MW, CN, AS, AVK, IH, and YaH processed the original observational datasets. YoH wrote the original draft of manuscript, and all coauthors contributed to review and editing.

*Competing interests.*  The authors declare that they have no conflict of interest.

*Acknowledgements.*  The authors are thankful to Editor Dr. Martina Krämer, and reviewers Drs. Minghui Diao and Emma Järvinen for their comments that contributed to improving the manuscript. This work was supported by the National Science Foundation (Award Numbers: 1213311 and 1842094). Observational data are provided by NCAR/EOL under the sponsorship of the National Science Foundation (https://data.eol.ucar.edu/, last access: 26 May 2020). The authors are grateful to NCAR's Data Support Section for providing ERA5 reanalysis data (https://doi.org/10.5065/BH6N-5N20, last access: 13 October 2019). The authors acknowledge high-performance computing support
from Cheyenne (https://doi.org/10.5065/D6RX99HX) provided by NCAR's Computational and Information Systems Laboratory, sponsored by the National Science Foundation. NCAR is sponsored by the National Science Foundation. Some of the computing for this project was performed at the University of Oklahoma (OU) Supercomputing Center for Education and Research (OSCER). The discussions of HIWC conditions and IKP with Walter Strapp, the discussions of secondary ice productions with Vaughan T. J. Phillips, and the discussions of model comparison with Lulin Xue are greatly appreciated. Major North American funding for flight campaigns was provided by the FAA William
Hughes Technical Center and Aviation Weather Research Program, the NASA Aeronautics Research Mission Directorate Aviation Safety Program, the Boeing Co., Environment and Climate Change Canada, the NRC of Canada, and Transport Canada. Major European campaign and research funding was provided by (i) the European Commission Seventh Framework Program in research, technological development and demonstration under grant agreement n°ACP2-GA-2012-314314, (ii) the European Aviation Safety Agency (EASA) Research Program under service contract n° EASA.2013.FC27. Further funding was provided by the Ice Crystal Consortium.

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

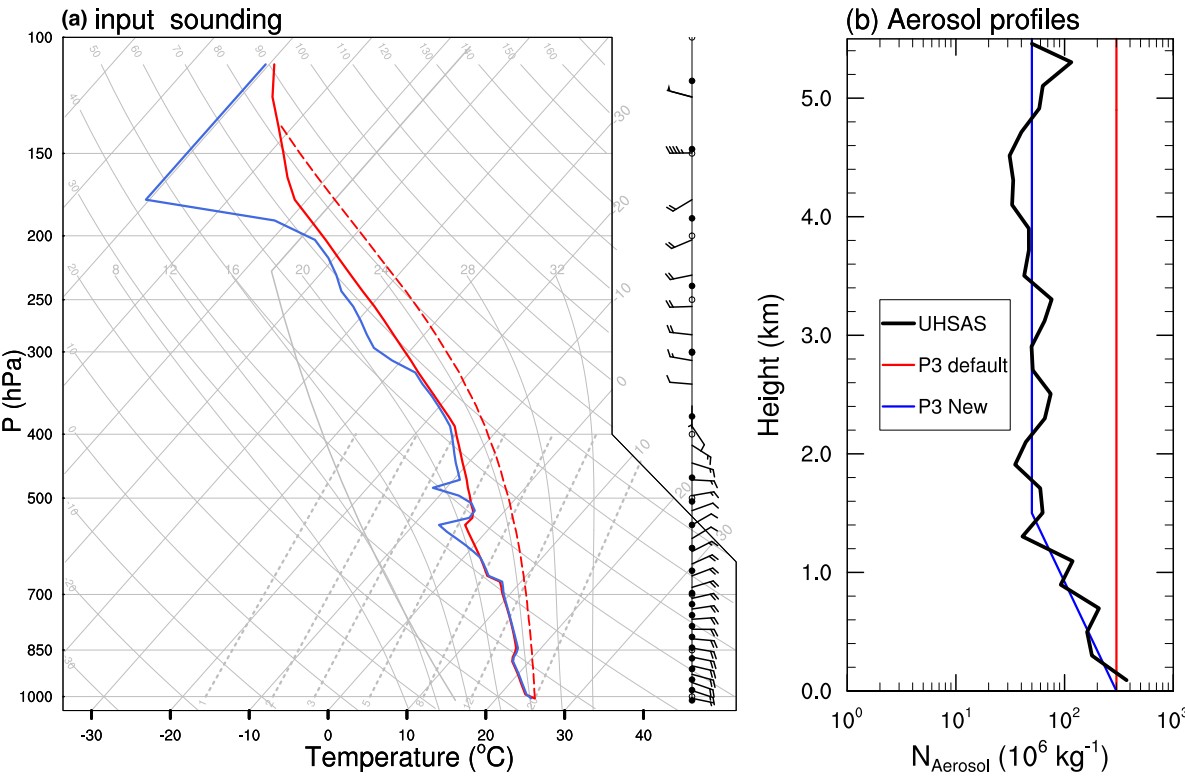

**Figure 1.** (a) Input sounding from the radiosonde released at Cayenne at 00:00 UTC 26 May 2015. One full wind barb represents 10 knot ($\sim$5.14 m s$^{-1}$). (b) Profiles of aerosol number mixing ratio (N$_{Aerosol}$ in units of $10^6$ kg$^{-1}$; UHSAS observation: black, default profile in P3 scheme: red; new profile based on UHSAS observation: blue).

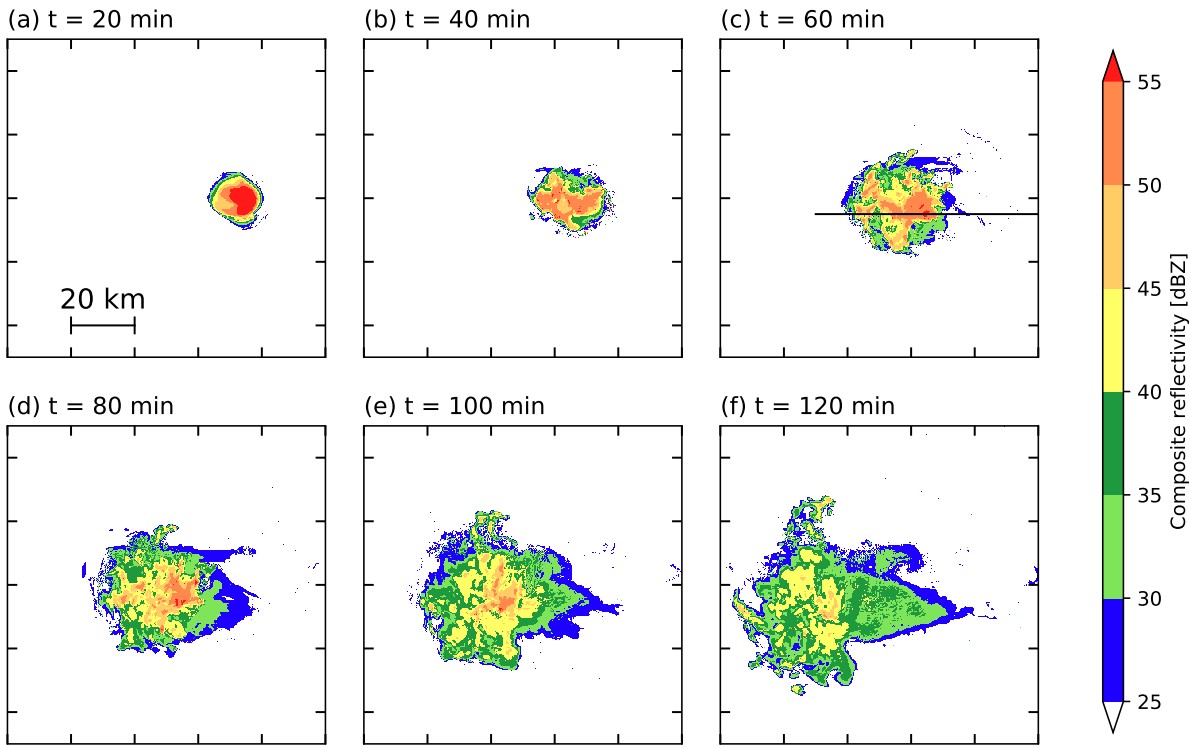

**Figure 2.** Composite reflectivity (dBZ) in SIPs250m at (a–f) $t = 20$, 40, 60, 80, 100, and 120 min, respectively. The black solid line in (c) indicates the location of cross section shown in Fig. 7. Tick marks are included every 20 km.

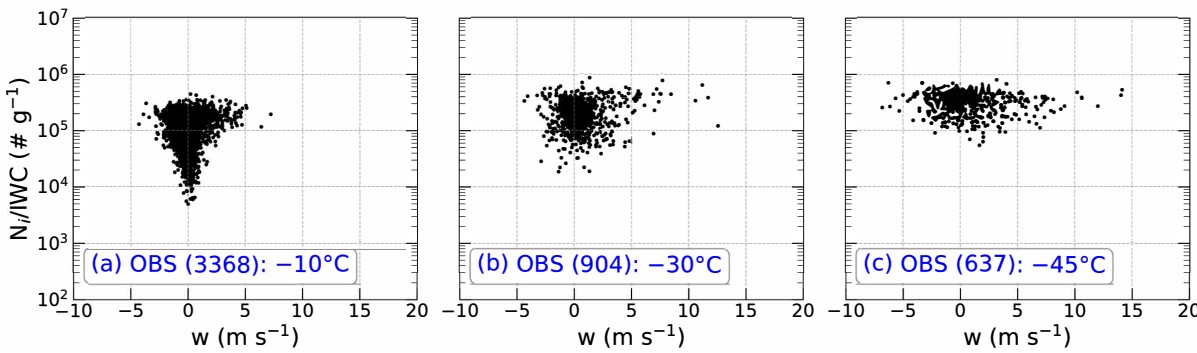

**Figure 3.** Scatter plots of observed ice number concentration (Ni, $\#\,\mathrm{m}^{-3}$) divided by ice water content (IWC, $\mathrm{g\,m}^{-3}$) (denoted as Ni/IWC) as a function of vertical velocity (w, $\mathrm{m\,s}^{-1}$) in regions with IWC $> 1\,\mathrm{g\,m}^{-3}$ from all flights during the Cayenne field campaign at temperatures of (a) $-10\,^{\circ}\mathrm{C}$, (b) $-30\,^{\circ}\mathrm{C}$, and (c) $-45\,^{\circ}\mathrm{C}$, and the observed temperature ranges of samples at the three levels are $-12.9$ to $-7.3\,^{\circ}\mathrm{C}$, $-33.0$ to $-27.3\,^{\circ}\mathrm{C}$, $-45.4$ to $-42.4\,^{\circ}\mathrm{C}$, respectively. The numbers in parentheses represent the total number of samples.

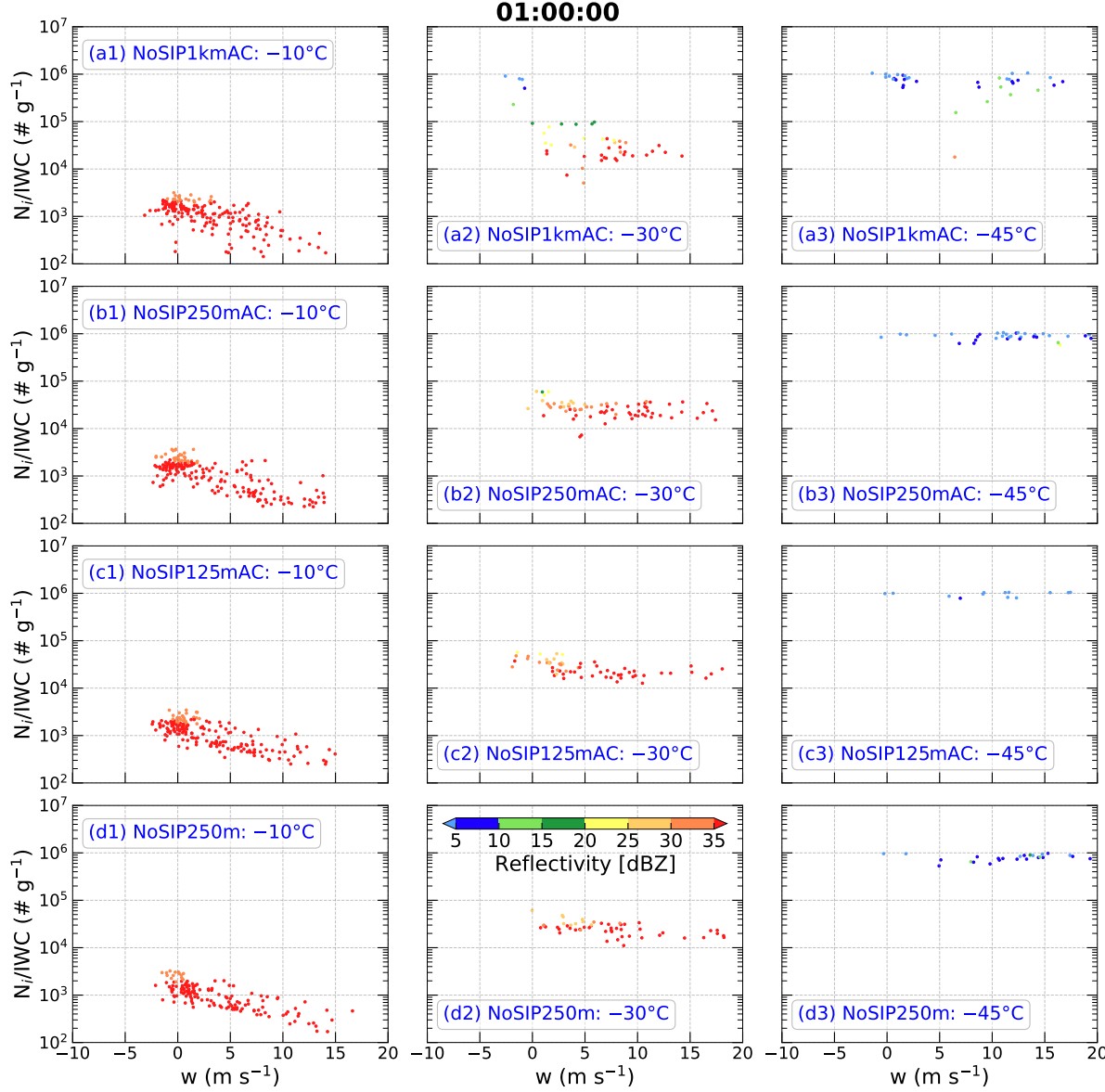

**Figure 4.** Scatter plots of simulated ice number concentration (Ni, # m$^{-3}$) divided by ice water content (IWC, g m$^{-3}$) (denoted as Ni/IWC) as a function of vertical velocity (w, m s$^{-1}$) in regions with IWC > 1 g m$^{-3}$ at temperatures of (left column) $-10°$C, (middle column) $-30°$C, and (right column) $-45°$C at $t = 60$ min in experiments (a1–a3) NoSIP1kmAC, (b1–b3) NoSIP250mAC, (c1–c3) NoSIP125mAC, and (d1–d3) NoSIP250m, respectively. The simulations at the three temperature levels are interpolated from the model outputs. The simulations with horizontal grid spacing < 1 km have been coarsened to 1 km for comparison by spatially averaging with a window size of 1 km × 1 km. The points are color-coded according to the magnitude of radar equivalent reflectivity factor (dBZ).

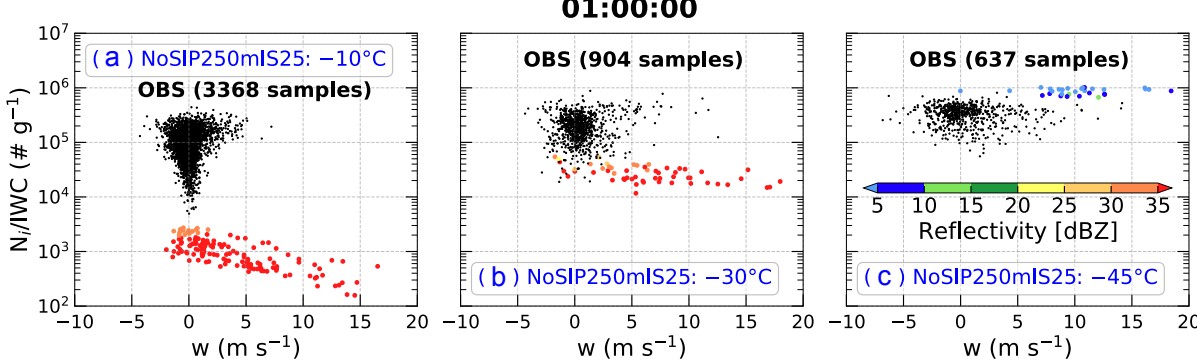

**Figure 5.** As Fig. 4 but for the experiment NoSIP250mIS25 overlaid with observations in Fig. 3.

**01:00:00**

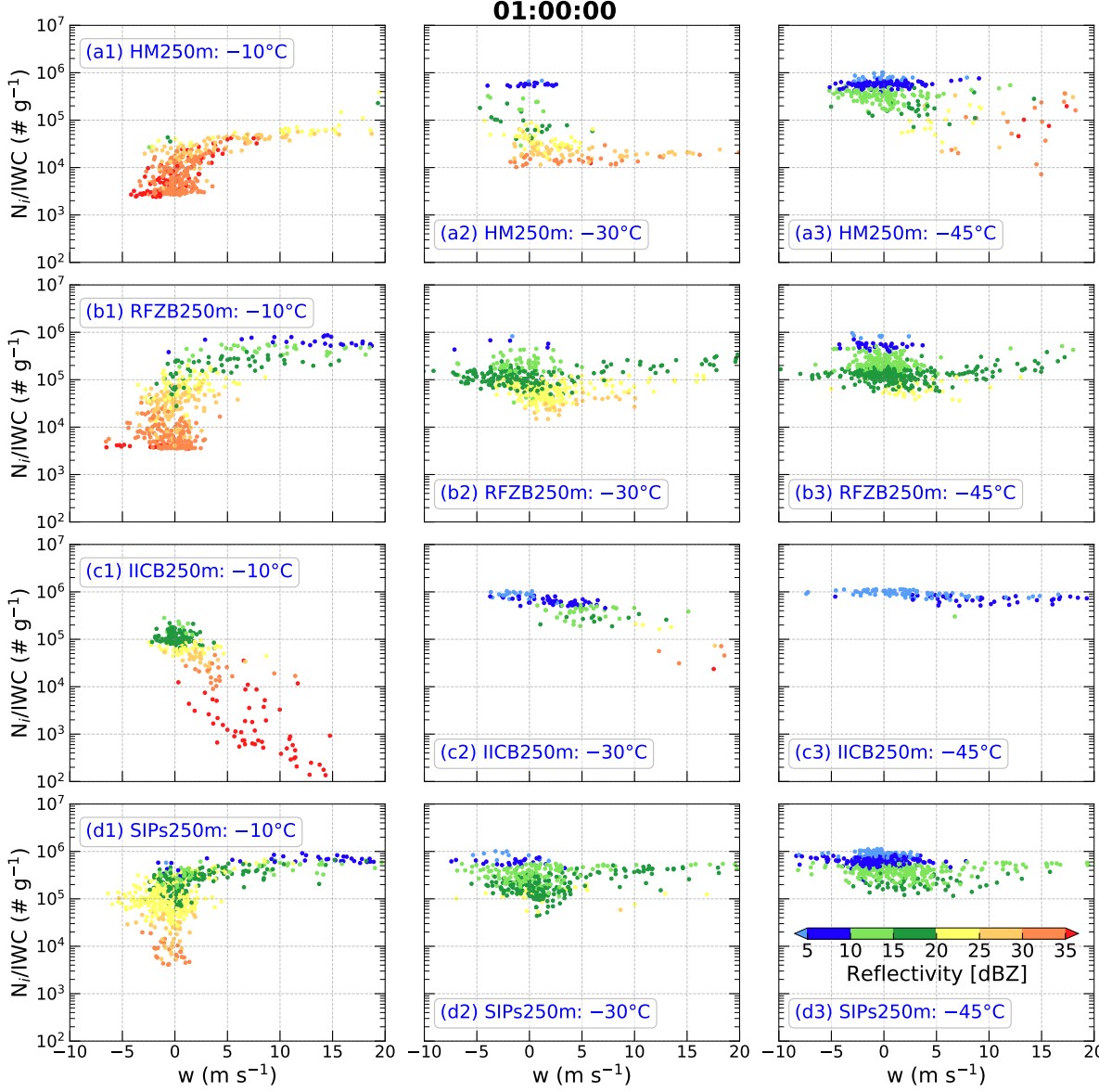

**Figure 6.** As Fig. 4 but for experiments (a1–a3) HM250m, (b1–b3) RFZB250m, (c1–c3) IICB250m, and (d1–d3) SIPs250m, respectively. The acronyms HM, RFZB, IICB and SIPs represent Hallett–Mossop process, raindrop freezing breakup, ice-ice collision breakup, secondary ice production processes, respectively.

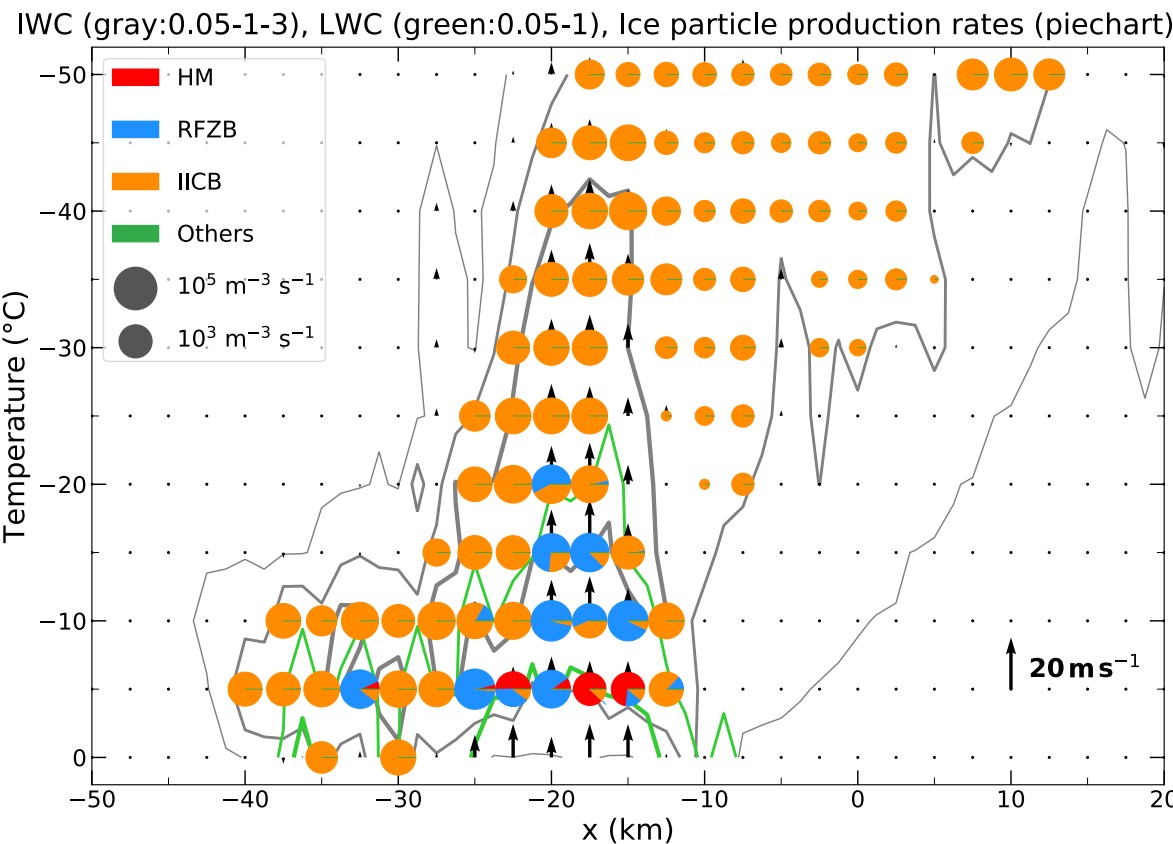

**Figure 7.** Vertical cross section along the line aligned along X shown in Fig. 2c of IWC (gray contours: 0.05, 1, and 3 g m$^{-3}$ from thin to the thick), LWC (green contours: 0.05, and 1 g m$^{-3}$ from thin to the thick), vertical velocity (vertical vectors), and the microphysical process rates relevant for ice particle production processes including (red) H-M mechanism, (blue) shattering of freezing droplets, (orange) fragmentation of ice–ice collision, and (green) other microphysical processes (i.e., ice nucleation, homogeneous and heterogeneous freezing of cloud droplets and rain) in regions with IWC > 1 g m$^{-3}$ at different temperatures in SIPs250m at $t = 60$ min. The pie charts denote ice particle production rates summed over all the source terms with the area of each color proportional to the ice particle production rate.

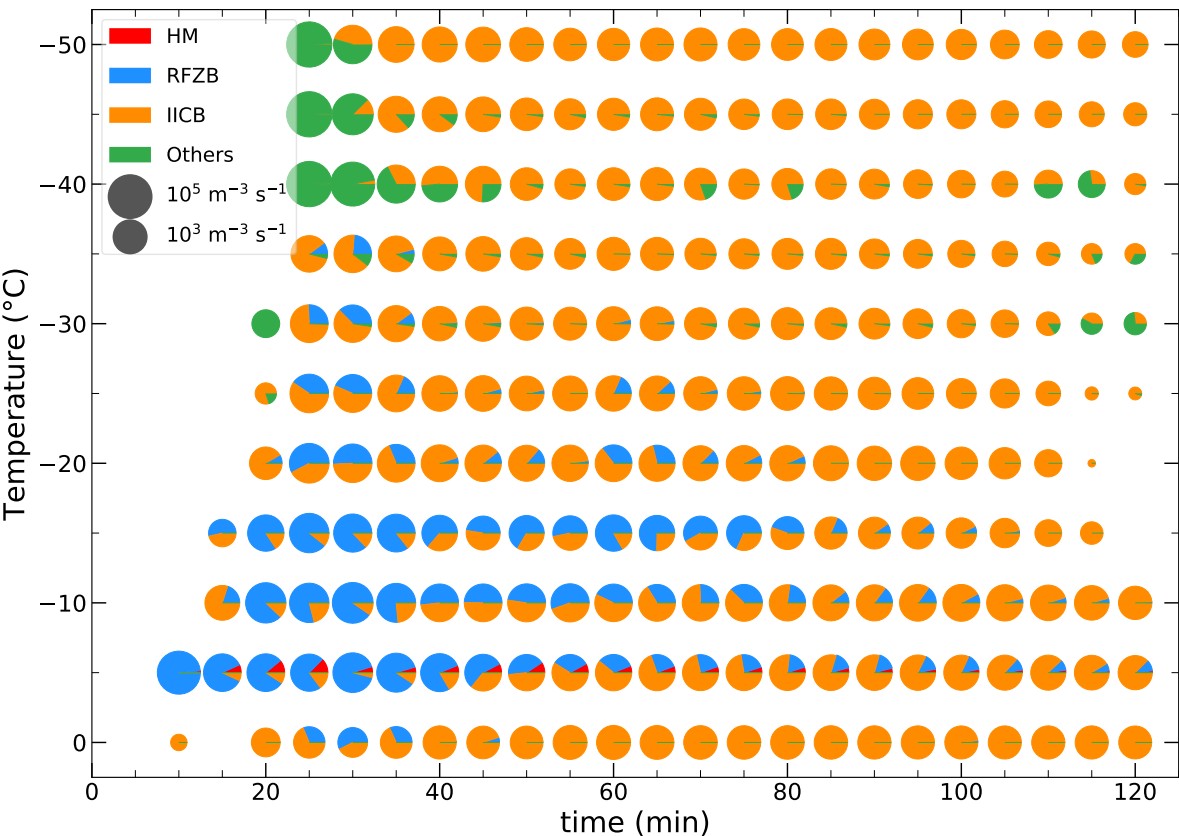

**Figure 8.** Time evolution of the averaged microphysical process rates relevant for ice particle production processes including (red) H-M mechanism, (blue) shattering of freezing droplets, (orange) fragmentation of ice–ice collision, and (green) other microphysical processes (i.e., ice nucleation, homogeneous and heterogeneous freezing of cloud droplets and rain) in regions with IWC > 1 g m$^{-3}$ at different temperatures. The pie charts denote ice particle production rates summed over all the source terms with the area of each color proportional to the ice particle production rate.

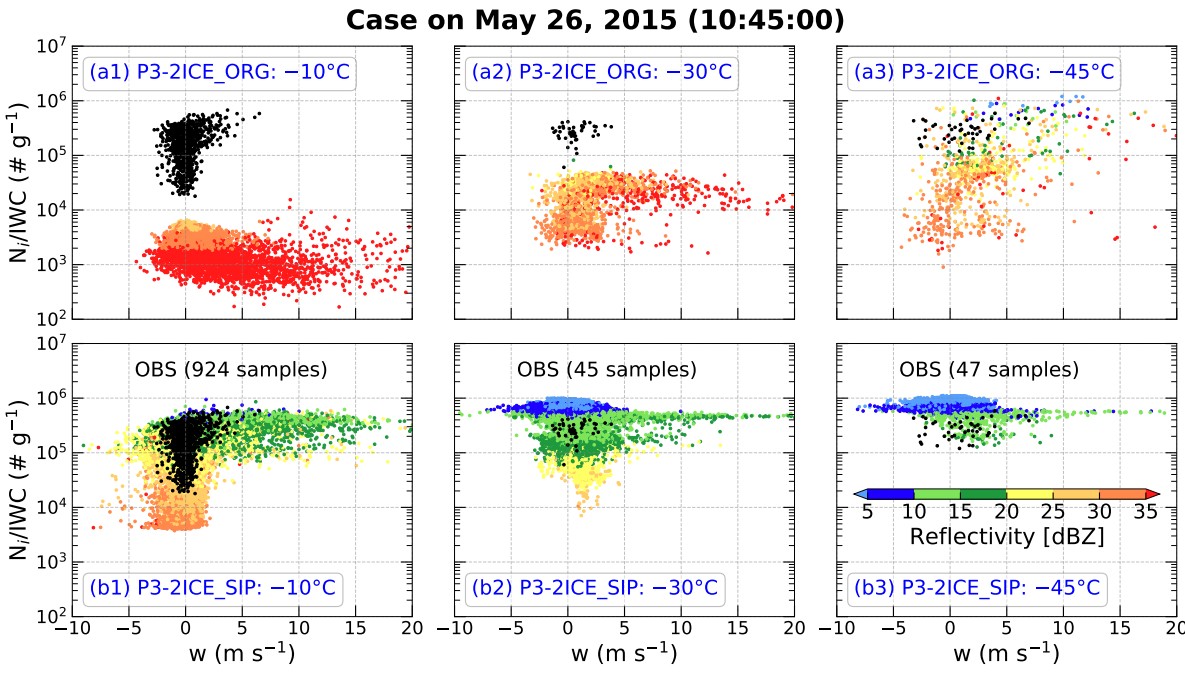

**Figure 9.** Scatter plots of (black) observed and (colorized) simulated ice number concentration (Ni, # m$^{-3}$) divided by ice water content (IWC, g m$^{-3}$) (denoted as Ni/IWC) as a function of vertical velocity (w, m s$^{-1}$) in regions with IWC $> 1$ g m$^{-3}$ at temperatures of (a1, b1) $-10°$C, (a2, b2) $-30°$C, and (a3, b3) $-45°$C. The observations are from the two flights, SAFIRE Falcon 20 and NRC Convair 580 (shown in Fig. 1 of Huang et al. (2021)), during the Cayenne field campaign, and the observed temperature ranges of samples at the three levels are $-12.6$ to $-7.9°$C, $-30.4$ to $-29.7°$C, $-44.7$ to $-43.6°$C, respectively. The simulation using the original P3 two-ice category configuration is shown in a1–a3, and the simulation using the P3 two-ice category configuration including the three SIP processes is shown in b1–b3. The simulations at the three temperature levels are interpolated from the 1-km model outputs. The scatters of simulations are color-coded according to the magnitude of radar equivalent reflectivity factor (dBZ).

**Table 1.** Sensitivity experiments

| Exp | dx, dy (m) | Aerosol profile | SIP processes |
|---|---|---|---|
| NoSIP1kmAC | 1000 | Constant | None |
| NoSIP250mAC | 250 | Constant | None |
| NoSIP150mAC | 125 | Constant | None |
| NoSIP250m | 250 | Observation | None |
| HM250m | 250 | Observation | H-M mechanism |
| RFZB250m | 250 | Observation | Raindrop freezing shattering |
| IICB250m | 250 | Observation | Fragmentation during ice-ice collision |
| SIPs250m | 250 | Observation | All SIP processes on |

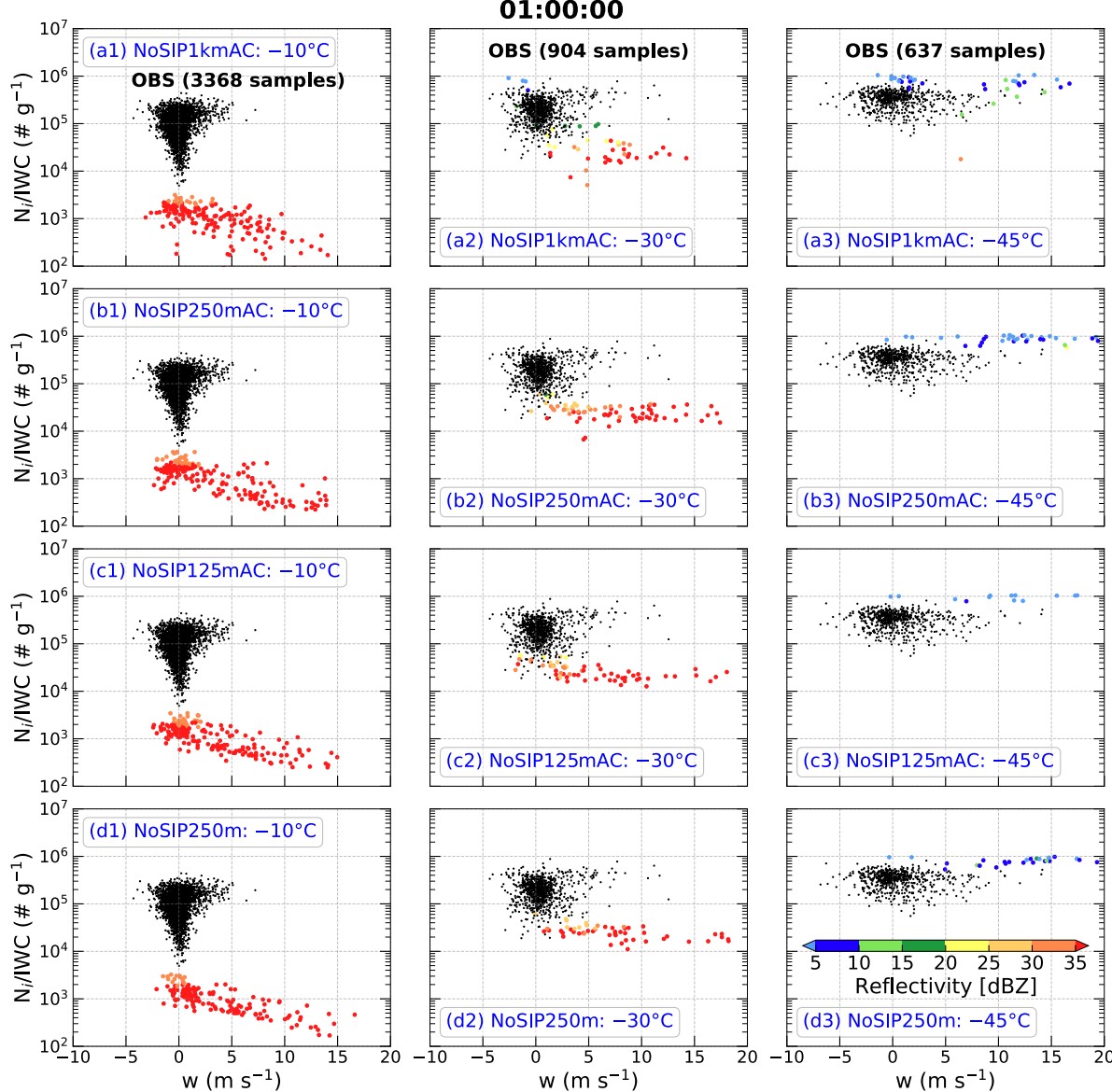

**Figure A1.** As Fig. 4 but overlaid with observations in Fig. 3.

**01:00:00**

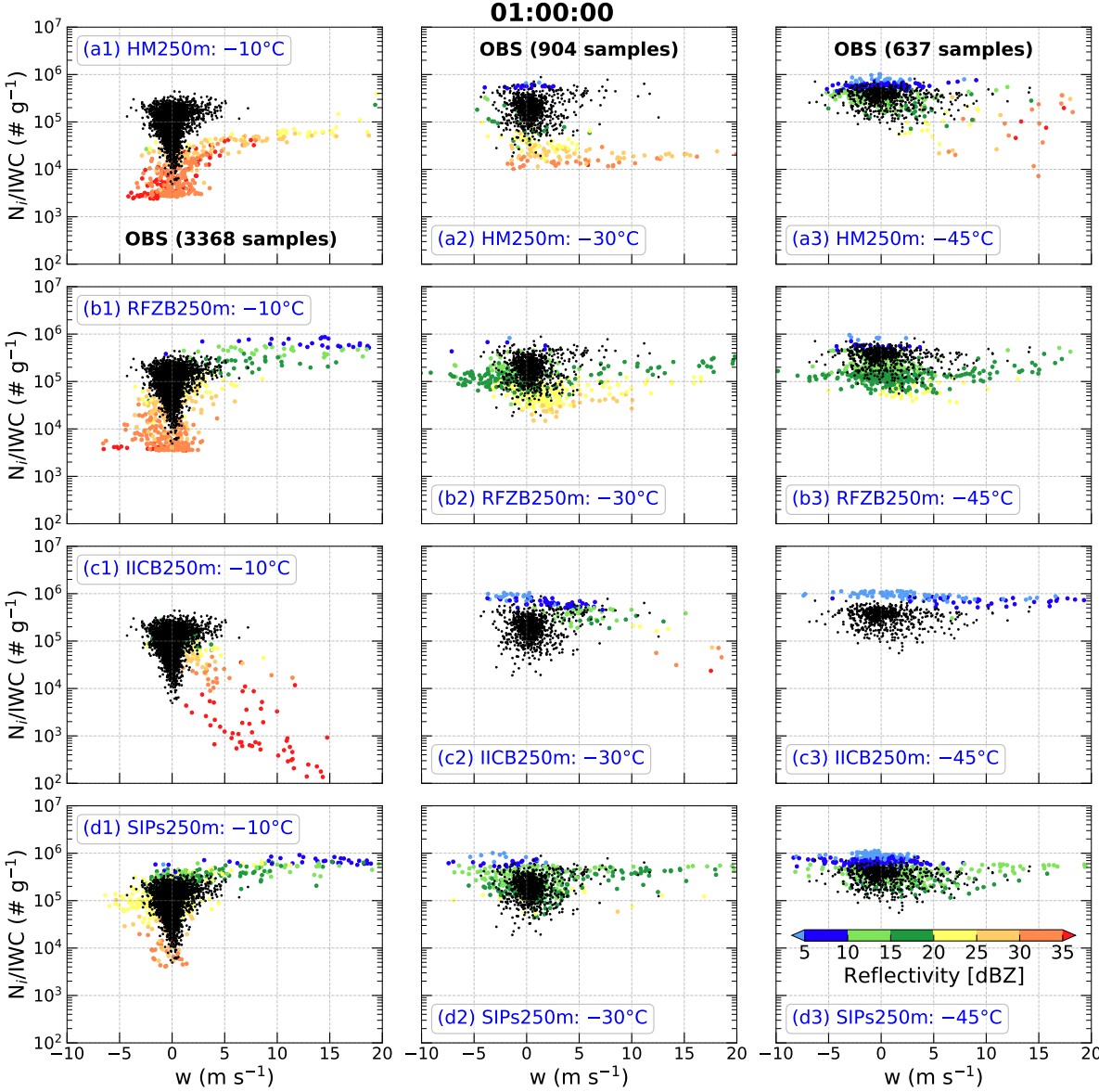

**Figure A2.** As Fig. 6 but overlaid with observations in Fig. 3.