# Peer review of "Microphysical processes producing high ice water contents (HIWCs) in tropical convective clouds during the HAIC-HIWC field campaign: dominant role of secondary ice production"

_Atmospheric Chemistry and Physics, 2021_

## Author Comment (AC1)

**Responses to the comments of Referee #1**

Referee #1 (Dr. Minghui Diao): In this work, the authors use observations from the High Altitude Ice Crystals (HAIC)-HIWC experiment to evaluate WRF model simulations based on the P3 cloud microphysics scheme. Specifically, model performance for representing high ice crystal number concentrations (Nice) observed during convective activity is examined. Several model setups are investigated, including the default P3 microphysics scheme, P3 scheme with various horizontal grid spacings, and P3 scheme that includes several secondary ice production (SIP) processes, such as (1) the rime-splintering or Hallett–Mossop (HM) process, (2) ice–ice collision fragmentation (IICB), (3) raindrop freezing breakup (RFZB). The results show that the P3 scheme that includes all three SIP mechanisms (HM, RFZB, and IICB) provides the most comparable results to the observed Nice / ice water content (IWC) ratio and radar reflectivity. The dominant SIP mechanism at various temperature ranges is also evaluated. Overall, the manuscript is well written. The model experimental design is straightforward and easy to follow. The reviewer recommends the manuscript being considered for publication after making the following revisions.

**Response:** We would like to express our acknowledgement for your efforts and constructive comments. Our point-by-point responses are given below. For convenience, the reviewer's comments are in **black** fonts, and our point-by-point responses are in **blue**. The line numbers in the response are based on the track-change manuscript.

**Major comments:**

(1) The sensitivity tests conducted for WRF mainly focus on the inclusion of various SIP processes. There are other factors that potentially affect the distribution of IWC and Nice significantly, such as the threshold of relative humidity with respect to ice (RHice) used to initiate ice nucleation. As far as the reviewer knows, P3 scheme still uses the ice nucleation formulation of Cooper (1986), similar to Morrison double moment scheme. If so, the reviewer thinks that the P3 scheme may be subject to similar problems seen in previous WRF simulations of convective systems.

Previously, D'Alessandro et al. (2017) evaluated several double moment schemes (i.e., Morrison 2-moment, Thompson 2-moment, and Thompson-Eidhammer aerosol aware scheme) in WRF model, and Diao et al. (2017) evaluated the Morrison 2-moment scheme in the NCAR CM1 cloud-resolving model (similar to WRF model). Both studies showed that the Cooper parameterization used to initiate ice nucleation has a RHice threshold that is too low

(D'Alessandro et al., 2017, doi:10.1002/2016JD025994; Diao et al., 2017, doi:10.1175/JAS-D-16-0356.1). When the Cooper parameterization is used in Morrison 2-moment, it does not allow clear-sky ice supersaturation to exceed 108%. When comparing WRF simulations against aircraft observations of anvil clouds during the NSF DC3 field campaign, such activation of ice nucleation at 108% leads to an underestimation of the occurrence frequency of ice supersaturation in both clear-sky and in-cloud conditions. Consequently, it allows ice nucleation to happen too early when RHice is still relatively low and leads to higher ice water content and ice crystal number concentrations than those seen in the observations.

The reviewer suggests testing the activation of Cooper 1986 parameterization at a higher ice supersaturation threshold (such as RHice = 125% or 130%). This would allow more clear-sky ice supersaturation to exist without turning into ice crystals too early. It would be interesting to see the impacts of such revision compared with the inclusion of SIP processes.

**Response:** Yes, the P3 scheme uses the ice nucleation formulation of Cooper (1986), and the ice supersaturation threshold is 5% (RHice = 105%) in the scheme. At the reviewer's suggestion, we conducted a sensitivity experiment using a higher ice supersaturation threshold of 25% (RHice = 125%, referred to as NoSIP250mIS25). From Fig. R1-1, it is seen that changing the ice supersaturation threshold does not influence the conclusions attained in this study. We have discussed this point in the revised manuscript.

Lines 281-292 in the track-change manuscript:

"Previous studies (e.g., D'Alessandro et al., 2017; Diao et al., 2017) showed that the ice supersaturation threshold in the ice nucleation parameterization of Cooper (1986) used in common microphysics schemes (e.g., Morrison et al., 2005; Morrison and Milbrandt, 2015) is too low, which can affect the distribution of ice water content and ice number concentration substantially. To examine the impact of varying this threshold, a sensitivity study changing the ice supersaturation threshold from 5% to 25% in the ice nucleation parameterization of the P3 scheme was conducted. The simulation is the same as NoSIP250m but the ice supersaturation threshold of 25% used in the ice nucleation parameterization (referred to as NoSIP250mIS25). Figure 5 shows scatter plots of simulated Ni/IWC for 0.1 mm $< D_{max} <$ 12.845 mm as a function of vertical velocity in regions with IWC $> 1$ g m$^{-3}$ linearly interpolated to the temperatures of $-10$, $-30$, and $-45$ °C at $t = 60$ min in NoSIP250mIS25 overlaid with observations in Fig. 3. From Fig. 5, the results in NoSIP250mIS25 are very similar to those in NoSIP250m (Figs. 4d1–d3), in terms of orders of magnitude of Ni/IWC and intensity of radar reflectivity at the three

temperature levels. It indicates that changing ice supersaturation threshold in the ice nucleation parameterization does not influence the conclusions attained in this study."

[Figure]

**Fig. R1-1.** Scatter plots of (black) observed and (colorized) simulated ice number concentration (Ni, # m$^{-3}$) divided by ice water content (IWC, g m$^{-3}$) (denoted as Ni/IWC) as a function of vertical velocity (w, m s$^{-1}$) in regions with IWC > 1 g m$^{-3}$ at temperatures of (left column) $-10$ °C, (middle column) $-30$ °C, and (right column) $-45$ °C at t = 60 min in experiments (a1–a3) NoSIP250m, (b1–b3) NoSIP250mIS25, and (c1–c3) SIPs250m, respectively. The simulations at the three temperature levels are interpolated from the model outputs. The simulations with horizontal grid spacing < 1 km have been coarsened to 1 km for comparison by spatially averaging with a window size of 1 km × 1 km. The points are color-coded according to the magnitude of radar equivalent reflectivity factor (dBZ).

(2) Following the first comment, evaluations of thermodynamic conditions (such as RHice) would be helpful besides the evaluation of Nice and IWC in relation to vertical velocity. It would be valuable to show the distributions of ice microphysical properties in relation to RHice

with or without SIP processes included as well as how these simulated distributions compare with observations.

**Response:** Our previous observational study (Hu et al. 2021) indicated that Ni and IWC are significantly correlated with environmental temperature and vertical velocity. We also tried examining the dependence on relative humidity, however, we found the quality of the observed relative humidity data was too poor to be used for our study. Therefore, we mainly focus on conditions of environmental temperature and vertical velocity in this study. Based on the response to the first comment, we can expect that relative humidity is not the main cause of the orders of magnitude differences in ice number concentrations between the experiments with and without SIP mechanisms.

Reference:

Hu, Y., McFarquhar, G.M., Wu, W., Huang, Y., Schwarzenboeck, A., Protat, A., Korolev, A., Rauber, R.M. and Wang, H., 2021: Dependence of Ice Microphysical Properties On Environmental Parameters: Results from HAIC-HIWC Cayenne Field Campaign. Journal of the Atmospheric Sciences, 78, 2957–2981. https://doi.org/10.1175/JAS-D-21-0015.1.

(3) The current study includes three SIP mechanisms, while previous literature pointed out other existing SIP mechanisms as well. If the current three SIP mechanisms already provide a similar amount of Nice as the observed value, wouldn't adding other SIP mechanisms in the future lead to Nice that is too high compared with observations? The reviewer suggests adding some discussions on this potential problem when more SIP processes are included in the WRF model.

**Response:** There are other SIP mechanisms discussed in the review paper (Korolev and Leisner, 2020) and mentioned in the introduction of this manuscript, such as fragmentation of sublimating ice particles, ice particle fragmentation due to thermal shock, and activation of INPs in transient supersaturation around freezing drops. However, to the best of our knowledge, currently there is only one recent attempt to parameterize fragmentation of sublimating ice particles (Deshmukh et al., 2021) and there are no parameterizations for the other two SIP mechanisms. T4herefore, we have not tested these SIP mechanisms. It should be noted that different SIP mechanisms have different conditions in which they efficiently operate, such as different environmental temperatures, existence of drops, and ice particle sizes. There might also exist competition between different SIP mechanisms, such as two mechanisms requiring the involvement of raindrops. Therefore, adding other SIP mechanisms would not necessarily lead to higher Ni. We have discussed these points in the revised manuscript.

Lines 108-117 in the track-change manuscript:

"There are other SIP mechanisms reviewed by Korolev and Leisner (2020) that are not considered in the simulations presented here, such as fragmentation of sublimating ice particles, ice particle fragmentation due to thermal shock, and activation of INPs in transient supersaturation around freezing drops. However, to the best of our knowledge, currently there is only one recent attempt to parameterize fragmentation of sublimating ice particles (Deshmukh et al., 2021) and there are no parameterizations for the other two SIP mechanisms, so it is difficult to implement them in simulations. It also should be noted that different SIP mechanisms operate efficiently in different conditions, which are functions of environmental temperature, existence of drops, and ice particle sizes, etc. Further, there can be competition between different SIP mechanisms operating at similar conditions, such as two mechanisms requiring the involvement of raindrops. Therefore, adding other SIP mechanisms would not necessarily lead to higher ice number concentration."

Reference:

Deshmukh, A., V. T. J. Phillips, A. Bansemer, S. Patade, and D. Waman: New Empirical Formulation for the Sublimational Breakup of Graupel and Dendritic Snow. Journal of the Atmospheric Sciences, https://doi.org/10.1175/JAS-D-20-0275.1, 2021.

Korolev, A. and Leisner, T.: Review of experimental studies of secondary ice production, Atmospheric Chemistry and Physics, 20, 11 767– 11 797, https://doi.org/10.5194/acp-20-11767-2020, 2020.

(4) Since Nice is a key observed variable used to evaluate WRF simulations, what is the range of uncertainty associated with Nice, as it is affected by various potential problems such as shattering and poorly defined depth of field? Can the authors give a range of observed Nice at various temperatures by using a more rigorous versus a less rigorous quality control procedure?

**Response:** We have discussed the uncertainty of ice particle size distribution in the companion paper (Huang et al. 2021): "The size distribution data with uncertainty of 10 %–100 % (Baumgardner et al., 2017) are processed following the general approach described in McFarquhar et al. (2017), with only center-in particles accepted, and corrections for out-of-focus particles (Korolev, 2007), shattered particles (Field and Heymsfield, 2003; Field et al., 2006; Korolev and Field, 2015), and particles partially within the photodiode array applied (Heymsfield and Parrish, 1978). Due to a poorly defined depth of field for small particles (Baumgardner et al., 2012) and the potential of shattered artifacts only n(D) for $D_{max} > 50$ μm are considered here." Unfortunately, the raw data files from HAIC/HIWC are not available to

us due to HAIC/HIWC data policy, so we cannot perform sensitivity tests to assess impact of different processing routines on derived properties. However, we would like to point out that both errors in particle sizing and concentration are size dependent. Maximum errors are associated with the images < 5 pixels (e.g., 50 µm for 2DS). For particles with $D_{max}$ < 50 µm maximum error in sizing due to the diffraction effects does not exceed a factor of 2. In other words, the measured sizes can be overestimated, and the average error in sizing is less than factor of 2. The calculated concentration of particles with $D_{max}$ < 50 µm can be overestimated due to the residual shattering artifacts by factor of 2 (i.e., the actual concentration for $D_{max}$ < 50 µm would be 2 times lower). The uncertainty in concentration due to the depth of field definition is expected to be within a factor of ±1.5. For particles with $D_{max}$ > 50 µm the uncertainty in particle sizing and concentration are expected to be less that those declared above. Given orders of magnitude differences in Ni in some of the plots, a factor of two is not important and our findings are robust. To examine the uncertainty of Ni, we have done some sensitivity tests where different lower limits (i.e., 50, 100, and 200 µm) are used. The conclusions are consistent among these sensitivity tests (Figs. R1-2–7). We have mentioned these results in our revised manuscript and added the related figures as a supplement.

Lines 228-237 in the track-change manuscript:

"Composite particle size distributions were derived from the Two Dimensional Stereo Imaging Probe (2D-S) and the Precipitation Imaging Probe (PIP) for the particles with $D_{max}$ between 0.01 and 12.845 mm. The observed Ni only considers contributions from ice crystals with $D_{max}$ > 0.05 mm due to the potential of shattered artifacts and small and poorly defined depth of field for small particles (Huang et al., 2021; Hu et al., 2021). There is considerable uncertainty in estimating concentrations of ice crystals with $D_{max}$ < 0.2 mm from current probes and processing algorithms (McFarquhar et al., 2017; O'shea et al., 2021). To examine the sensitivity of findings to ice crystal concentrations in small sizes, sensitivity tests using different lower limits of $D_{max}$ (i.e., 0.05, 0.1, and 0.2 mm) were conducted. The qualitative findings are consistent among these sensitivity tests (Figs. S1–S6), so only results using the lower limit of $D_{max}$ = 0.1 mm are discussed here. More details on the processing and uncertainty of observations can be found in Huang et al. (2021) and Hu et al. (2021)."

Reference:

Huang, Y., Wu, W., McFarquhar, G. M., Wang, X., Morrison, H., Ryzhkov, A., Hu, Y., Wolde, M., Nguyen, C., Schwarzenboeck, A., et al.: Microphysical processes producing high ice water contents (HIWCs) in tropical convective clouds during the HAIC-HIWC field

campaign: evaluation of simulations using bulk microphysical schemes, Atmospheric Chemistry and Physics, 21, 6919–6944, https://doi.org/10.5194/acp-21-6919-2021, 2021.

McFarquhar, G. M., and Coauthors, 2017: Processing of ice cloud in situ data collected by bulk water, scattering, and imaging probes: Fundamentals, uncertainties, and efforts towards consistency. Ice Formation and Evolution in Clouds and Precipitation: Measurement and Modeling Challenges, Meteor. Monogr., No. 58, Amer. Meteor. Soc., 11.1–11.33, https://doi.org/10.1175/AMSMONOGRAPHS-D-16-0007.1.

[Figure]

**Fig. R1-2.** As Fig. R1-1 but for experiments NoSIP1kmAC, NoSIP250mAC, NoSIP125mAC, and NoSIP250m for 50 μm < $D_{max}$ < 12845 μm, respectively.

[Figure]

**Fig. R1-3.** As Fig. R1-1 but for experiments NoSIP1kmAC, NoSIP250mAC, NoSIP125mAC, and NoSIP250m for 100 μm < $D_{max}$ < 12845 μm, respectively.

[Figure]

**Fig. R1-4.** As Fig. R1-1 but for experiments NoSIP1kmAC, NoSIP250mAC, NoSIP125mAC, and NoSIP250m for 200 μm < $D_{\max}$ < 12845 μm, respectively.

[Figure]

**Fig. R1-5.** As Fig. R1-1 but for experiments HM250m, RFZB250m, IICB250m, and SIPs250m for 50 μm < $D_{max}$ < 12845 μm, respectively.

[Figure]

**Fig. R1-6.** As Fig. R1-1 but for experiments HM250m, RFZB250m, IICB250m, and SIPs250m for 100 μm < $D_{max}$ < 12845 μm, respectively.

[Figure]

**Fig. R1-7.** As Fig. R1-1 but for experiments HM250m, RFZB250m, IICB250m, and SIPs250m for 200 μm < $D_{max}$ < 12845 μm, respectively.

(5) Can the authors elaborate on how liquid droplets or raindrops are separated from ice crystals in 2DS and PIP measurements? The paper of Huang et al. (2021) briefly mentioned that "The two optical array probes, 2D-S and PIP, recorded 2D images of ice crystals nominally in the size range of 10–1280 and 100–6400 μm, respectively." But this statement is not entirely true because 2D-S and PIP can capture liquid phase as well.

**Response:** We discussed the detection of liquid cloud drops in detail in our observational study (Hu et al. 2021). Cloud segments, where the frequency of Rosemount Icing Detector on the Falcon 20 was decreasing and was lower than 40 kHz (Mazin et al. 2001) or where Nt detected

by CDP-2 was larger than 10 cm$^{-3}$ (Lance et al. 2010; Ding et al. 2020), identified the presence of liquid water content and were removed. A frequency threshold of 39.7 kHz was applied to the different model of the Rosemount Icing Detector installed on the Convair-580. We have mentioned this point in the revised manuscript. It would be extremely unlikely that there would be larger drops present in the 2DS/PIP if no small liquid drops were detected by the Rosemount Icing Detector and CDP-2 (this would probably only happen if the aircraft flew through rain beneath cloud base). Nevertheless, to ensure that such time periods were not present we inspected 2DS and PIP images to verify that there were no periods where exclusively round particles, indicative of liquid, were present.

Lines 226-228 in the track-change manuscript:

"Cloud segments with the presence of liquid water were identified from voltage changes of the Rosemount Icing Detector and from the total concentration measured by the Cloud Droplet Probe version 2 (CDP-2), and not considered in this analysis."

Reference:

Ding, S., G. M. McFarquhar, S. W. Nesbitt, R. J. Chase, M. R. Poellot, and H. Wang, 2020: Dependence of mass—Dimensional relationships on median mass diameter. Atmosphere, 11, 756, https://doi.org/10.3390/atmos11070756.

Hu, Y., McFarquhar, G.M., Wu, W., Huang, Y., Schwarzenboeck, A., Protat, A., Korolev, A., Rauber, R.M. and Wang, H., 2021: Dependence of Ice Microphysical Properties on Environmental Parameters: Results from HAIC-HIWC Cayenne Field Campaign. Journal of the Atmospheric Sciences, 78, 2957–2981. https://doi.org/10.1175/JAS-D-21-0015.1.

Lance, S., C. A. Brock, D. Rogers, and J. A. Gordon, 2010: Water droplet calibration of the Cloud Droplet Probe (CDP) and in-flight performance in liquid, ice and mixed-phase clouds during ARCPAC. Atmos. Meas. Tech., 3, 1683–1706, https://doi.org/10.5194/amt-3-1683-2010.

Mazin, I. P., A. V. Korolev, A. Heymsfield, G. A. Isaac, and S. G. Cober, 2001: Thermodynamics of icing cylinder for measurements of liquid water content in supercooled clouds. J. Atmos. Oceanic Technol., 18, 543–558, https://doi.org/10.1175/1520-0426(2001)018<0543:TOICFM>2.0.CO;2.

(6) The analysis uses the size range (Dmax) of 0.1 to 12.845 mm, which is slightly higher than the range of 0.05 – 12.845 mm used in the companion paper (Huang et al., 2021). Is there a reason that a higher minimum threshold of Dmax (0.1 mm) is used in this work?

In addition, is the model output of ice microphysical properties (e.g., IWC, Nice, mass-mean diameter, etc.) re-calculated based on this partial size range in order to match the size range of the observations? A partial size selection procedure has been applied to model output as shown in previous studies, e.g., Fridlind et al. (2007), Eidhammer et al. (2014), Patnaude et al. (2021) https://doi.org/10.5194/acp-21-1835-2021, Yang et al. (2020) DOI: 10.1002/essoar.10504450.1.

Since the smaller ice particles dominate the Nice value, if the authors only evaluate the range of 0-0.05 mm (given that the observations at this range have large uncertainties), do the simulations show too many or too few ice particles at this size range? This size range (0-0.05 mm) is quite important because if the simulations already provide several orders of magnitude of higher Nice than observed Nice at 0-0.05 mm, getting a more similar Nice for > 0.05 mm by adding SIP processes can potentially lead to worse results when considering the entire size range of ice crystals. Alternatively, if the default P3 simulations provide smaller Nice at 0-0.05 mm compared with observations, it can be interpreted as either the observed Nice is overestimated due to shattering, or the default P3 simulations really underestimate Nice even at small ice crystal sizes.

**Response:** There is a possibility that particles with maximum dimensions less than 200 μm are overestimated by some processing algorithms (McFarquhar et al. 2017). Thus, we examined the sensitivity of our results to the use of different lower limits of ice crystal sizes (i.e., 50, 100, and 200 μm), which is also included in the companion paper (Huang et al., 2021). From the results shown in Figs. R1-2–7, the conclusions are consistent regardless of the lower threshold used, so we only discussed the results using minimum $D_{max}$ threshold of 100 μm in detail in the manuscript. We have mentioned the sensitivity tests in our revised manuscript and added the related figures as supplement.

Yes, we attained the ice number distribution function from the model and re-calculated the related variables for the same range of $D_{max}$ as was done for the observations (i.e., 0.1–12.845 mm). This procedure has been applied in the companion paper (Huang et al., 2021). Therefore, the comparison between simulations and observations is consistent.

Due to uncertainties in the size-dependent depth of field, it is not really possible to calculate the concentrations of ice crystals with $D_{max}$ < 50 μm from the 2D-S. Nevertheless, we did the comparison for $D_{max}$ < 50 μm that the reviewer suggested. Figures R1-8–9 show the results at the full observed range of $D_{max}$ (10–12845 μm) including ice crystals for $D_{max}$ < 50 μm. As with the other comparisons, Ni in the simulations without including SIP mechanisms are significantly underestimated compared to the observations, especially at temperatures of −10

and −30 °C (Fig. R1-8). The simulation including all the three SIP mechanisms produces Ni closest to the observations, although it slightly underestimates the Ni, especially at temperatures of −30 and −45 °C (Fig. R1-9 d1–d3). This bias may result from an overestimate of observed Ni due to shattering or uncertainty due to the depth of field, or the underestimate of simulated Ni for $D_{max}$ < 50 μm in P3 scheme.

Lines 228-237 in the track-change manuscript:

"Composite particle size distributions were derived from the Two Dimensional Stereo Imaging Probe (2D-S) and the Precipitation Imaging Probe (PIP) for the particles with $D_{max}$ between 0.01 and 12.845 mm. The observed Ni only considers contributions from ice crystals with $D_{max}$ > 0.05 mm due to the potential of shattered artifacts and small and poorly defined depth of field for small particles (Huang et al., 2021; Hu et al., 2021). There is considerable uncertainty in estimating concentrations of ice crystals with $D_{max}$ < 0.2 mm from current probes and processing algorithms (McFarquhar et al., 2017; O'shea et al., 2021). To examine the sensitivity of findings to ice crystal concentrations in small sizes, sensitivity tests using different lower limits of $D_{max}$ (i.e., 0.05, 0.1, and 0.2 mm) were conducted. The qualitative findings are consistent among these sensitivity tests (Figs. S1–S6), so only results using the lower limit of $D_{max}$ = 0.1 mm are discussed here. More details on the processing and uncertainty of observations can be found in Huang et al. (2021) and Hu et al. (2021)."

[Figure]

**Fig. R1-8.** As Fig. R1-1 but for experiments NoSIP1kmAC, NoSIP250mAC, NoSIP125mAC, and NoSIP250m for 10 μm < $D_{max}$ < 12845 μm, respectively.

[Figure]

**Fig. R1-9.** As Fig. R1-1 but for experiments HM250m, RFZB250m, IICB250m, and SIPs250m for 10 μm < $D_{max}$ < 12845 μm, respectively.

**Minor comments:**

Line 19 – 22, some of the temperature ranges do not have the complete range. See suggestions in brackets: "… shattering of freezing droplets dominates ice particle production in HIWC regions at temperatures > −15C [temperatures between -15C and 0C??] during the early stage of convection, and fragmentation during ice–ice collisions dominates at temperatures > −15C [temperatures between -15C and 0C??] during the later stage of convection and at temperatures < −20C [temperatures between -40C and -20C??] over the whole convection period."

**Response:** Revised according to the suggestions.

Line 218, "... interpolated to temperatures of -10, -30, …", can the author explain what kind of interpolation was done?

**Response:** We used linear interpolation in the vertical direction in our study. We have indicated it in the revised manuscript.

Figures 6 and 7 have two colors that are very similar in the legend, that is, HM in red and Others in fuchsia. Even though the colors are readable from a computer screen, when printed out on paper these two colors look almost the same. The reviewer suggests changing "Others" to some other colors, such as green, yellow or light gray.

**Response:** The color has been updated.

**Responses to the comments of Referee #2**

Referee #2 (Dr. Emma Järvinen): In this study, simulations of tropical deep convective clouds performed with the Weather Research and Forecasting (WRF) model are evaluated against observations from the High Altitude Ice Crystals (HAIC)-HIWC experiment. As a companion paper to Huang et al., 2021, a closer look on the role of secondary ice production (SIP) to generate observed HIWC regions is taken. Three SIP mechanisms are investigated: the Hallett–Mossop (H-M) process, ice–ice collision fragmentation (IICB) and raindrop freezing breakup (RFZB). It is found that simulations including all three SIP processes successfully produces HIWC regions in all three temperature levels that were investigated. The results highlight the importance of SIP processes in controlling the ice water content in the studied tropical convective clouds. The paper is well written and the model experiments are easy to follow. There are only minor shortcomings that should be addressed before publication. First, the ice crystal observations would have deserved more discussion and ideally, the authors could have included their best estimation of the magnitude of the uncertainties related to these observations. Furthermore, the uncertainties related to the existing SIP parameterisations should be highlighted more. After these minor revisions the manuscript is recommended for publication.

**Response:** We would like to express our acknowledgement for your efforts and constructive comments. Our point-by-point responses are given below. For convenience, the reviewer's comments are in **black** fonts, and our point-by-point responses are in **blue**. The line numbers in the response are based on the track-change manuscript.

**General comments**

1. For evaluation of the role of SIP it is crucial to have reliable observations of ice crystal number concentrations (Ni). However, the authors do not discuss the Ni observations during the HAIC-HIWC field campaign or, more importantly, their uncertainties. Although Huang et al., 2021 contains information of cloud microphysical observations, the relevant measurement methods should be summarised also in this manuscript. For example, Huang et al. (2021) states that the Ni measurements were derived from the Two Dimensional Stereo Imaging Probe (2D-S) and the Precipitation Imaging Probe (PIP) for the size range for Dmax>50 µm but in this manuscript use a higher lower size limit of 100 µm. This choice should be discussed.

    **Response:** We have summarized the relevant measurement methods and uncertainties in the revised manuscript. There are some studies that suggest that particles less than 200 µm

may be overestimated by some processing algorithms (McFarquhar et al. 2017). To examine the dependence of our results on the lower size threshold used to calculate Ni, we have conducted sensitivity tests where different lower size limits (i.e., 50, 100, and 200 µm) were used to calculate Ni. The conclusions are consistent among these sensitivity tests, so only the results using the lower limit of 100 µm are discussed in the manuscript. We have mentioned these points in our revised manuscript and added the related figures as supplement.

Lines 228-237 in the track-change manuscript:

"Composite particle size distributions were derived from the Two Dimensional Stereo Imaging Probe (2D-S) and the Precipitation Imaging Probe (PIP) for the particles with $D_{max}$ between 0.01 and 12.845 mm. The observed Ni only considers contributions from ice crystals with $D_{max} > 0.05$ mm due to the potential of shattered artifacts and small and poorly defined depth of field for small particles (Huang et al., 2021; Hu et al., 2021). There is considerable uncertainty in estimating concentrations of ice crystals with $D_{max} < 0.2$ mm from current probes and processing algorithms (McFarquhar et al., 2017; O'shea et al., 2021). To examine the sensitivity of findings to ice crystal concentrations in small sizes, sensitivity tests using different lower limits of $D_{max}$ (i.e., 0.05, 0.1, and 0.2 mm) were conducted. The qualitative findings are consistent among these sensitivity tests (Figs. S1–S6), so only results using the lower limit of $D_{max} = 0.1$ mm are discussed here. More details on the processing and uncertainty of observations can be found in Huang et al. (2021) and Hu et al. (2021)."

Reference:

McFarquhar, G. M., and Coauthors, 2017: Processing of ice cloud in situ data collected by bulk water, scattering, and imaging probes: Fundamentals, uncertainties, and efforts towards consistency. Ice Formation and Evolution in Clouds and Precipitation: Measurement and Modeling Challenges, Meteor. Monogr., No. 58, Amer. Meteor. Soc., 11.1–11.33, https://doi.org/10.1175/AMSMONOGRAPHS-D-16-0007.1.

2. How is the model sampled in order to get Ni and IWC values in the size range of 0.1-12.845 mm? How is the ice particle size defined in the model? What is the possible error in Ni if the model and observations have a different definition for size?

**Response:** We attained the ice number distribution function from the model and re-calculated the related variables for the same range and same bin size of $D_{max}$ as the observed.

This procedure has been applied in the companion paper (Huang et al., 2021). Therefore, the comparison between simulations and observations is consistent.

3. The sensitivity of the model to horizontal resolution and aerosol profile is discussed in Sec. 3.2. The results are discussed in terms of Ni/IWC values but it is not well justified why the 250m-resolution model was chosen for the sensitivity studies including SIP processes.

**Response:** As discussed in Section 3.2, we see that the simulated results using different horizontal resolution are similar. Thus, the simulation bias in Ni/IWC does not mainly result from the model resolution. As discussed in Section 2.2.2, "Lebo and Morrison (2015) found overall storm characteristics had limited sensitivity when horizontal grid spacing was decreased below 250 m in their simulated squall lines." A simulation using 125-m grid spacing consumes much more computing resources than a simulation using 250-m grid spacing. Therefore, we chose 250-m grid spacing for the sensitivity studies including SIP processes. We have indicated this point in the revised manuscript.

Lines 212-215 in the track-change manuscript:

"A horizontal grid spacing of 250 m and the more realistic vertical profile of aerosol number mixing ratio are chosen for the sensitivity experiments including SIP processes, because results reveal the model resolution and aerosol profile are not the main source of model biases in simulating HIWCs (discussed in detail in Section 3.2), and because a simulation using 125-m grid spacing consumes much more computing resources than a simulation using 250-m grid spacing."

Reference:

Lebo, Z. and Morrison, H.: Effects of horizontal and vertical grid spacing on mixing in simulated squall lines and implications for convective strength and structure, Monthly Weather Review, 143, 4355–4375, https://doi.org/10.1175/MWR-D-15-0154.1, 2015.

4. Fragmentation of ice in ice-ice collisions is shown to dominate the ice particle production rates outside the updraft core region even at temperatures warmer than -15°C. Do the authors consider this as a realistic outcome or a result of the way ice-ice collisions were implemented in the model?

Laboratory studies suggest that the number of ice ejected in collisions has a strong dependence of temperature with a maximum around -18°C (Takahashi, Nagao & Kushiyama, 1995). The break-up of ice crystals in ice-ice collisions is explained by the different ice crystal surface properties (brittle surface with plate-like growth or dendrites

colliding with compact graupel). Plate-like and dendritic growth contributing to fragmentation is taking place in the temperature region between -15°C and -20°C, which explains the observed maximum in ice production rate in the laboratory studies. Ice crystals around -5 and -10°C have more compact columnar shapes. Is this difference in shapes taken into account in the parameterisation?

**Response:** In this study, we adopted a physically-based parameterization of ice multiplication by breakup during ice–ice collision proposed by Phillips et al. (2017). This parameterization scheme is based on an energy conservation principle, in which the number of new fragments per collision is dependent on cloud species (i.e., hail, graupel, snow or crystals whether dendritic or spatial planar), collision kinetic energy, temperature, and colliding particles' size and rimed fraction. Parameters in the scheme are estimated based on previous laboratory experiment by Takahashi et al. (1995) and field observations by Vardiman (1978). Therefore, this parameterization scheme is more reasonable than other schemes that are only temperature-dependent and just simply fit the ice particle production rate to the results attained during graupel–graupel collisions in the laboratory experiment by Takahashi et al. (1995). Moreover, fragmentation during ice–ice collision is not only dependent on ice number ejected in the collision of colliding pair but also dependent on the ice particle number concentration. It means that higher fragmentation rate of colliding pair does not mean higher total fragmentation of ice in ice-ice collisions, and vice versa. We agree that there might exist an overproduction of ice due to ice–ice collisional breakup parameterization, which was also mentioned in previous studies. In this study, we used current commonly accepted microphysical parameterizations but acknowledge that there are inevitably uncertainties in the representation of actual physical processes. The community has a general consensus that some parameterizations including fragmentation during ice–ice collision have to be revisited, and they require feedback from more theoretical studies, field campaigns including remote-sensing and in-situ observations and laboratory studies. The most important outcome of this paper is that one of the necessary conditions for HIWC formation in MCSs is enhanced production of secondary ice. The question about the actual mechanisms of SIP and the associated rates of ice production is beyond the scope of this study. These points have been discussed in the revised manuscript. Lines 152-156 in the track-change manuscript:

"In this study, current commonly accepted microphysical parameterizations are used. However, there are uncertainties in the parameterization of both primary ice production and SIP mechanisms (Korolev and Leisner, 2020). In fact, uncertainties in the parameterization

of primary ice production also transfer to uncertainties in SIP processes. Therefore, more theoretical studies, field campaigns including remote-sensing and in-situ observations, and laboratory studies should be conducted to constrain parameterizations of both primary ice production and SIP mechanisms in the future (Morrison et al., 2020)."

Reference:

Phillips, V. T., Yano, J.-I., and Khain, A.: Ice multiplication by breakup in ice–ice collisions. Part I: Theoretical formulation, Journal of the Atmospheric Sciences, 74, 1705–1719, https://doi.org/10.1175/JAS-D-16-0224.1, 2017.

Takahashi, T., Y. Nagao, and Y. Kushiyama, 1995: Possible high ice particle production during graupel–graupel collisions. J. Atmos. Sci., 52, 4523–4527, https://doi.org/10.1175/1520-0469(1995)052,4523:PHIPPD.2.0.CO;2.

Vardiman, L., 1978: The generation of secondary ice particles in clouds by crystal–crystal collision. J. Atmos. Sci., 35, 2168–2180, https://10.1175/1520-0469(1978)035,2168:TGOSIP.2.0.CO;2.

5. It is important to state that there are severe uncertainties in the parameterisations of SIP mechanisms. Although there is a statement about this in Summary and Conclusions, this could be also stated earlier in the manuscript (e.g. in Sec. 2.1). Ideally, some additional sensitivity tests by tuning the different SIP parameterisations could have been performed.

**Response:** We have stated the uncertainties in the parameterizations of SIP mechanisms in Section 2.1. As the response to the comment #4, in this study, we used current commonly accepted microphysical parameterizations although they have uncertainties in the representation of physical processes. The community has a general consensus that some parameterizations including fragmentation during ice–ice collision have to be revisited, and they require feedback from more theoretical studies, field campaigns including remote-sensing and in-situ observations and laboratory studies. The most important outcome of this paper is that one of the necessary conditions for HIWC formation in MCSs is enhanced production of secondary ice. The question about the actual mechanisms of SIP and the associated rates of ice production is beyond the scope of this study. These points have been discussed in the revised manuscript.

Lines 152-156 in the track-change manuscript:

"In this study, current commonly accepted microphysical parameterizations are used. However, there are uncertainties in the parameterization of both primary ice production and SIP mechanisms (Korolev and Leisner, 2020). In fact, uncertainties in the parameterization

of primary ice production also transfer to uncertainties in SIP processes. Therefore, more theoretical studies, field campaigns including remote-sensing and in-situ observations, and laboratory studies should be conducted to constrain parameterizations of both primary ice production and SIP mechanisms in the future (Morrison et al., 2020)."

Lines 432-437 in the track-change manuscript:

"For example, the high ice production rate due to the fragmentation during ice-ice collisions is highly uncertain, and its high production rate in anvil cloud regions between −40 °C and −50 °C (Fig. 7) is rarely seen in observations. However, these uncertainties do not influence the main conclusions due to the orders of magnitude differences in ice number concentrations between the experiments with and without SIP mechanisms. This study enhances understanding of the processes leading to formation of the numerous small crystals in HIWC regions in which enhanced production of secondary ice is one of the necessary conditions."

**Minor comments**

Line 203: Can the authors explain the choice to discuss the results in the form of Ni/IWC? What is the additional value in this representation?

**Response:** Because IWC varies among samples and there may exist bias in the simulated IWC compared to observations, using the form of Ni/IWC allows us to focus on ice number distribution. Further, because most observed and simulated IWC are between 1 and 3 g m$^{-3}$, and only very small instances of IWC greater than 3 g m$^{-3}$ occur, the results using either Ni or Ni/IWC are very similar.

Figure 4: Can the observations be added to the corresponding subplots or even combine Figures 3 and 4?

**Response:** If the observations are added to the corresponding subplots, they will mask some simulated samples. Instead, we add a figure combining Figs. 3 and 4 to the appendix (Fig. A1) for reference.

Figure 5: Same as for Fig. 4. It would be helpful to have the observations included. In addition, please explain the acronyms HM, RFZB, IICB and SIPs in the figure caption to improve readability.

**Response:** Same as the response to the last comment. Figure A2 including observations and simulations is shown in the appendix for reference. The acronyms HM, RFZB, IICB and SIPs have been explained in the figure caption.

Line 256: What kind of significance test was performed? Please add this information.
**Response:** We used *t*-test from SciPy package (https://docs.scipy.org/doc/scipy/reference/generated/scipy.stats.linregress.html). We have indicated this in the revised manuscript.

Line 345: Some discussion of how generating more ice in the early or lower parts of the cloud though SIP will lead to HIWC regions with more small ice crystal at colder and higher part of the cloud would be helpful to visualise the dynamics of these systems.
**Response:** We have added the related discussion in the revised manuscript.
Lines 388-389 in the track-change manuscript:
"More small ice particles generated in the early or lower level cloud through SIP processes also can increase the small ice crystals at upper cloud through vertical advection."